



# OH-chemistry of non-methane organic gases (NMOG) emitted from laboratory and ambient biomass burning smoke: evaluating the influence of furans and oxygenated aromatics on ozone and secondary NMOG formation.

Matthew M. Coggon[1,2], Christopher Y. Lim[3], Abigail R. Koss[1,2,‡], Kanako Sekimoto[1,2,4], Bin Yuan[1,2,†], Jessica B Gilman[2], David H. Hagan[3], Vanessa Selimovic[5], Kyle Zarzana[1,2], Steven S. Brown[2], James M. Roberts[2], Markus Müller[6], Robert Yokelson[5], Armin Wisthaler[7,8], Jordan E. Krechmer[9], Jose L. Jimenez[1,10], Christopher Cappa[11], Jesse Kroll[3], Joost de Gouw[1,10], and Carsten Warneke[1,2]

[1]Cooperative Institute for Research in Environmental Sciences, University of Colorado, Boulder, CO, USA
[2]NOAA Earth Systems Research Laboratory Chemical Sciences Division, Boulder, CO, USA
[3]Department of Civil and Environmental Engineering, Massachusetts Institute of Technology, Cambridge, MA, USA
[4]Graduate School of Nanobioscience, Yokohama City University, Yokohama, Kanagawa, Japan
[5]Department of Chemistry and Biochemistry, University of Montana, Missoula, MT, USA
[6]Ionicon Analytik, Innsbruck, Austria
[7]Institute for Ion Physics and Applied Physics, University of Innsbruck, Innsbruck , Austria
[8]Department of Chemistry, University of Oslo, Oslo, Norway
[9]Aerodyne Research, Inc., Billerica, MA, USA
[10]Department of Chemistry, University of Colorado, Boulder, CO, USA
[11]Department of Civil and Environmental Engineering, University of California, Davis, CA, USA
[‡]now Department of Civil and Environmental Engineering, Massachusetts Institute of Technology, Cambridge, MA, USA
[†]now at Institute for Environment and Climate Research, Jinan University, Guangzhou, China.

**Correspondence:** Matthew Coggon (matthew.m.coggon@noaa.gov), Carsten Warneke (carsten.warneke@noaa.gov)

**Abstract.**

Chamber oxidation experiments conducted at the Fire Sciences Laboratory in 2016 are evaluated to identify important chemical processes contributing to the OH chemistry of biomass burning non-methane organic gases (NMOG). Based on the decay of primary carbon measured by proton transfer reaction time-of-flight mass spectrometry (PTR-ToF-MS), it is confirmed

5  that furans and oxygenated aromatics are among the NMOG emitted from western United States fuel types with the highest reactivities towards OH. The oxidation processes and formation of secondary NMOG masses measured by PTR-ToF-MS and iodide clustering time-of-flight chemical ionization mass spectrometry (I-CIMS) is interpreted using a box model employing a modified version of the Master Chemical Mechanism (v. 3.3.1) that includes the OH oxidation of furan, 2-methylfuran, 2,5-dimethylfuran, furfural, 5-methylfurfural, and guaiacol. The model supports the assignment of major PTR-ToF-MS and

10  I-CIMS signals to a series of anhydrides and hydroxy furanones formed primarily through furan chemistry. This mechanism is applied to a Lagrangian box model used previously to model a real biomass burning plume. The updated mechanism reproduces the decay of furans and oxygenated aromatics and the formation of secondary NMOG, such as maleic anhydride. Based on model simulations conducted with and without furans, it is estimated that furans contributed up to 10% of ozone and over 90%





of maleic anhydride formed within the first 4 hours of oxidation. It is shown that maleic anhydride is present in a biomass
burning plume transported over several days, which demonstrates the utility of anhydrides as tracers for aged biomass burning
plumes.

## 1 Introduction

Biomass burning is a significant source of atmospheric non-methane organic gases (NMOG). Once emitted, biomass burn-
ing NMOG may undergo photochemical reactions to form ozone and secondary organic aerosol (SOA) (Hobbs et al., 2003;
Yokelson et al., 2009; Akagi et al., 2013). Wildfire smoke is believed to significantly contribute to summer-time ozone levels
in fire-prone regions, such as the western United States (Jaffe et al., 2008, 2013, 2018). An assessment of historical ozone
data from 1989 - 2004 has shown that daily mean ozone increases by 2 ppb for every 1 million acres of area burned (Jaffe
et al., 2008). A warming, drier climate is likely to increase fire activity, which may lead to increased ozone and PM2.5 levels
in susceptible regions (Westerling, 2006; Jaffe and Wigder, 2012; Brey and Fischer, 2016; Ford et al., 2018).

Despite its importance, the atmospheric chemistry of biomass burning smoke remains poorly-understood due to the com-
plexity of smoke processing. Field observations have shown that ozone enhancement ratios ($\Delta O_3/\Delta CO$) may increase (e.g.
0.7 ppb/ppb, Andreae et al., 1988; Mauzerall et al., 1998), decrease (e.g. -0.07 ppb/ppb, Alvarado et al., 2010), or remain
unchanged downwind of wildfires (Jaffe and Wigder, 2012). The extent of ozone production depends on multiple factors, in-
cluding NMOG/$NO_x$ ratios, downwind meteorology, and incident solar radiation (Akagi et al., 2013; Jaffe et al., 2018). Ozone
production may also be slowed through peroxyacetyl nitrate formation (PAN), which is affected, in part, by NMOG function-
ality and total $NO_x$ and NMOG emissions (Alvarado et al., 2010; Liu et al., 2016; Müller et al., 2016; Jaffe et al., 2018).
Biomass burning emissions are produced by distillation and pyrolysis, as well as glowing and flaming combustion (Yokelson
et al., 1996). Primary NMOG speciation is largely driven by pyrolysis temperatures and fuel composition (e.g., Sekimoto et al.,
2018; Hatch et al., 2015), whereas $NO_x$ emissions generally increase with increased flaming combustion and greater fuel
nitrogen content (Burling et al., 2010).

Secondary NMOG may provide insights into the chemical processes that contribute to smoke oxidation. Several studies have
identified important secondary NMOG formed from aging of biomass burning emissions (Yokelson et al., 2003; Müller et al.,
2016; Bruns et al., 2017; Hartikainen et al., 2018); however, the mechanisms that lead to secondary NMOG formation remain
unclear. For example, maleic anhydride has been identified as a significant secondary NMOG formed in smoke within hours of
oxidation (Müller et al., 2016; Bruns et al., 2017; Hartikainen et al., 2018). Maleic anhydride is a known product of aromatic
oxidation, but it also an end product of furan chemistry (Bierbach et al., 1995; Aschmann et al., 2011, 2014). Understanding
the pathways leading to secondary NMOG formation may be useful in constraining smoke properties (e.g. plume age), or
identifying significant ozone and SOA precursors.

Few studies have modeled the detailed chemical mechanisms leading to secondary NMOG formation (Mason et al., 2001;
Alvarado et al., 2015; Müller et al., 2016; Liu et al., 2016). This work is challenging because a large fraction (22 - 56%)
of the identified reactive carbon is associated with compounds whose OH chemistry is unknown or has not been specified





in atmospheric chemistry mechanisms (Hatch et al., 2017). These species could significantly contribute to ozone or SOA formation. For example, Alvarado et al. (2015) modeled the evolution of ozone and SOA formed downwind of a prescribed fire and found that $O_3$ production was strongly sensitive to the inclusion of unknown NMOG having OH rate constants of $\sim$
$10^{-11}$ cm$^3$molec$^{-1}$s$^{-1}$. Müller et al. (2016) explicitly modeled the oxidation of 16 NMOG emitted from a small understory fire using the Master Chemical Mechanism (MCM v. 3.3.1, Jenkin et al., 1997, 2003, 2015; Saunders et al., 2003). The model captured the loss of important reactive primary NMOG and reproduced the formation of $O_3$ and PAN. The formation of maleic anhydride could not be explained by the model, which reflects the need for additional mechanism development.

The studies described above demonstrate that highly reactive organic compounds play an important role in the OH oxidation
of young biomass burning plumes. Laboratory studies evaluating the reactivity of biomass burning NMOG have shown that furans, oxygenated aromatics, and aliphatic hydrocarbons (e.g. monoterpenes and cyclopentadiene) are major contributors to calculated or measured OH reactivity (Stockwell et al., 2015; Gilman et al., 2015; Hatch et al., 2015; Bruns et al., 2017; Hartikainen et al., 2018). Hartikainen et al. (2018), for example, found that furans and phenolic compounds were among the most reactive NMOG emitted from logwood emissions. The detailed chemical mechanisms of these compounds have been
studied in single-component systems (Bierbach et al., 1995; Alvarez et al., 2009; Aschmann et al., 2011, 2014; Zhao and Wang, 2017; Yee et al., 2013; Lauraguais et al., 2014; Finewax et al., 2018); however, these mechanisms have not been widely implemented into models of biomass burning smoke. Müller et al. (2016) included simple mechanisms for furan and furfural; however, other major furan species, such as 2-methylfuran, 2,5-dimethylfuran, and 5-methylfurfural, were omitted.

A few studies have evaluated biomass burning OH oxidation using proton-transfer-reaction time-of-flight mass spectrometry
(PTR-ToF-MS). PTR-ToF-MS is capable of measuring 50-80% of the primary NMOG mass emitted from biomass burning, including oxygenates, aromatics, and heterocyclic compounds (Hatch et al., 2017). The remaining mass includes compounds that are that are difficult to quantify by proton transfer, such as alkanes and small alkenes. The PTR-ToF-MS is effective in monitoring the evolution of some secondary NMOG, such as maleic anhydride (Müller et al., 2016). Nonetheless, limitations on isomeric specificity (Hatch et al., 2017), fragmentation, and sensitivity may hinder the ability of the PTR-ToF-MS to measure
other secondary oxygenates or multi-functionalized organics (Yuan et al., 2017). The iodide clustering time-of-flight chemical ionization mass spectrometer (I$^-$-ToF-CIMS, herinafter I-CIMS for brevity) is well-suited to measure oxygenated NMOG. The I-CIMS is sensitive to acids and multi-functional oxygenates (Lee et al., 2014), which are likely to form as secondary NMOG in biomass burning plumes. I-CIMS has been used to evaluate primary NMOG emissions (e.g., Priestley et al., 2018; Reyes-Villegas et al., 2018; Tomaz et al., 2018), biomass burning emissions aged by nocturnal processes (Ahern et al., 2018;
Reyes-Villegas et al., 2018), and primary particle-phase components (Gaston et al., 2016). I-CIMS spectra of the NMOG resulting from the OH oxidation of biomass burning smoke have yet to be reported.

Presented here are PTR-ToF-MS and I-CIMS measurements from chamber experiments conducted during the 2016 laboratory component of the Fire Influence on Regional to Global Environments and Air Quality Experiment (FIREX-AQ) conducted at the Fires Sciences Laboratory in Missoula, MT. Based on these data, modifications are made to the MCM (v.3.3.1) to in-
clude the reactions of highly reactive NMOG, including furan, furfural, 2-methylfuran, 2,5-dimethylfuran, 5-methylfurfural, and guaiacol. This mechanism is applied to a 0-D box model to interpret observed increases in secondary NMOG measured


by the PTR-ToF-MS and I-CIMS. The chemical mechanism is also applied to a Lagrangian model previously used to evaluate the OH chemistry of an ambient biomass burning plume (Müller et al., 2016). The model output is compared to measured secondary NMOG and $O_3$ production to evaluate the influence of furans and oxygenated aromatics on the chemistry of a real biomass burning plume.

## 2    Methods

### 2.1    Campaign description

The laboratory component of the FIREX-AQ intensive was conducted at the U.S. Forest Service Fire Sciences Laboratory in Missoula, MT during October - November 2016. The purpose of this study was to simulate the emissions and atmospheric oxidation of biomass burning smoke resulting from the combustion of western U.S. fuels. A full description of the campaign, experimental setup, types of fuels burned, and resulting emissions of key NMOG is provided elsewhere (Selimovic et al., 2018; Koss et al., 2018).

Burn experiments were conducted in a large, indoor combustion room as described by Selimovic et al. (2018). Inside the room, fuels were assembled on a bed centered below a 20 m (L) × 1.6 m (ID) exhaust stack. The fuels were ignited by a heating plate, and the resulting smoke was vented through the stack at a constant velocity of ∼ 3 m s$^{-1}$. Smoke was sampled by instrumentation on a platform located ∼17 m above of the fuel bed or directed to other areas of the laboratory through tubing and ductwork.

The fuels were chosen to represent ecosystems prone to wildfires in the western U.S and included components (i.e, leaves, stems, trunks, and duff) of the following species: ponderosa pine (*Pinus ponderosa*), lodegepole pine (*Pinus contorta*), Engelmann spruce (*Picea engelmanii*), Douglas fir (*Pseudotsuga menziesii*), subalpine fir (*Abies lasiocarpa*), manzanita (*Arctostaphylos*) and chamise (*Adenostoma fasciculatum*). The components of each fuel type were burned individually and in mixtures designed to mimic a real forested ecosystem. A full description of these fuels, including harvesting location, composition, and dry weight are provided elsewhere (Selimovic et al., 2018).

### 2.2    Chamber OH Oxidation Experiments

Semi-batch oxidation experiments were performed using an OH-oxidation chamber. The apparatus, subsequently referred to as the "mini chamber", consists of a 150 L Teflon bag centrally located between two UVC lamps (narrow peak emission 254 nm, Ultra-Violet Products, Inc.). OH is produced by the photolysis of ozone in the presence of water according to reactions R1 – R2

$$O_3 + h\nu \rightarrow O^1D \tag{R1}$$


$O^1D + H_2O \rightarrow 2OH$          (R2)

Other sources of OH, such as HONO and aldehydes, were also introduced into the chamber via injection of biomass burning smoke. The photon flux at the center of the chamber was measured to be $\sim 3 \times 10^{15}$ photons $cm^{-2}$ $s^{-1}$ using a photodiode sensor (Thorlabs S120VC). Previous studies using oxidation flow reactors have investigated the radical chemistry under 254 nm irradiation to quantify non-OH losses and $RO_2$ pathways (Peng et al., 2016; Peng and Jimenez, 2017; Peng et al., 2019). A

similar analysis is conducted here and discussed in Section 3.2.3.

A suite of particle and gas-phase instruments sampled from the bag through stainless-steel and Teflon tubing, respectively. This study focuses on gas-phase measurements; a complete description of aerosol measurements is provided by Lim et al. (2019).

The mini chamber was located in a room $\sim 30$ m from the top of the stack. To quickly deliver smoke to the chamber, stack

air was drawn through a 30 m L × 20 cm D aluminum duct. The residence time within the duct was < 2 s and particle losses were minor (Lim et al., 2019). Gas-phase species have a high affinity to metal surfaces and losses to the aluminum ductwork are possible (Deming et al., 2019). It is difficult to assess these losses quantitatively; however, the average NMOG/acetonitrile ratio measured in the mini chamber was only15% lower than that measured from the stack (see Supplemental Information for more details).

Mini chamber experiments were conducted in the following manner: (1) Prior to each burn, the bag was flushed with clean humidified air for $\sim 45$ min. (2) Immediately following fuel ignition, smoke diluted with clean air was injected into the bag at a $\sim 1:10$ ratio. Injection proceeded until the burn finished, or until particle concentrations maximized inside the mini chamber. For most experiments, the distribution of NMOG and particles represented an integrated sample of all phases of the burn. (3) 40 ppb of deuterated butanol (d-butanol, added to monitor OH exposure) was injected into the chamber and the particle and

gas mixture was allowed to mix for $\sim 5$-10 min. (4) A stream of clean, humidified air (RH $\sim 30$ %) doped with $\sim 70$ ppb ozone was continuously added to the mini chamber to match instrumentation sampling flows ($\sim 6$ L $min^{-1}$). (5) Once aerosol and gas-phase concentrations stabilized, the UVC lamps were turned on and photochemistry proceeded for $\sim 30$-45 min. (6) At the end of the experiment, the bag was flushed with clean dilution air in preparation for subsequent experiments.

Gas and particle-phase concentrations were corrected for dilution by monitoring the decay of acetonitrile, which is present at

high concentrations in biomass burning smoke, is slow to react with OH (lifetime $\sim$335 d at OH = $1.5 \times 10^6$ molec $cm^{-3}$), and is not significantly lost to Teflon surfaces (Krechmer et al., 2016; Pagonis et al., 2017). CO was not used as a dilution tracer due to observed CO production following the initiation of photochemistry. The dilution rate estimated by acetonitrile decay agreed well with flow rate calculations (Lim et al., 2019). OH exposures in the mini chamber were estimated based on the dilution-corrected loss of d-butanol ($k_{OH} = 3.4 \times 10^{-12}$ $cm^3$ $molec^{-1}$ $s^{-1}$, Barmet et al., 2012). For most primary NMOG, losses were

dominated by OH oxidation. For some species, photolysis played a significant role. Non-OH loss processes affecting important NMOG are discussed in Section 3.2.3.



## 2.3 Instrumentation

### 2.3.1 NMOG Measurements

NMOG measurements were conducted using a high-resolution proton transfer reaction time-of-flight mass spectrometer (PTR-
ToF-MS, Yuan et al., 2016) and an iodide clustering time-of-flight chemical ionization mass spectrometer (I-CIMS, Aero-
dyne/Tofwerk, AG). The instruments were deployed to measure primary smoke emissions from the stack and room, as well as
aged emissions from the mini chamber. During a burn (and while the mini chamber was filling with smoke), both instruments
sampled from the stack to characterize primary NMOG emissions. At the end of a burn, the sampling lines were switched
and the instruments sampled from the mini chamber through a 10 m long x 3 mm OD PFA inlet at a total flow rate of $\sim$ 3
L min$^{-1}$. A full description of the primary NMOG measurements is provided elsewhere and only a brief description of these
measurements are provided here (Koss et al., 2018).

The PTR-ToF-MS measured at 1 Hz to capture the decay of primary NMOG and formation of secondary species, respec-
tively. The drift tube was operated with an electric field to number density ratio (*E/N*) of 120 Td, and the high-resolution
mass spectrometer (max resolution $\sim$ 4500) scanned ions with *m/z* 12 - 500 Th. The mass spectrometer resolves the molec-
ular formulae of isobaric species, but cannot distinguish isomers. This presents challenges in reacting systems as secondary
NMOG formed by OH oxidation could have the same molecular formula as primary NMOG. Koss et al. (2018) identified the
distribution of primary NMOG during FIREX-AQ using gas chromatography pre-separation to measure isomer contributions.
For most fuel types, over 90% of the PTR-ToF-MS signal could be assigned. The primary groups detected by PTR-ToF-MS
were small oxygenates ($\sim$ 50% v/v, dominated by acetic acid, formaldehyde, methanol, and acetaldehyde), aromatics ($\sim$ 10%,
dominated by catechol, phenol, methoxy phenols, benzene, and toluene), furans ($\sim$ 10%, dominated by 5-methylfurfural, 2-
furfural, furanone, furan, and methyl + dimethyl furans) and a broad range of hydrocarbons ($\sim$ 15%, dominated by ethene,
propene, butene, and 1,3-butadiene). The same NMOG assignments and sensitivities are applied here to masses observed to be
enhanced prior to NMOG oxidation. The temporal profile of some larger primary oxygenates (C > 5) were also influenced by
the formation of secondary isomers or unidentified primary species (see Section 3.2.2). Calibration factors for primary species
are calculated from measured or estimated proton-transfer rate constants (uncertainty < 50%, Sekimoto et al., 2017).

Masses detected after the initiation of photochemistry are assigned based on previous literature observations and modeling
evidence (see Section 3.2.2). A number of laboratory and field studies employing PTR-ToF-MS and OP-FTIR have observed
formation of formic acid ($CH_2O_2-H^+$, *m/z* 47), acetic acid ($C_2H_4O_2-H^+$, *m/z* 61), maleic anhydride ($C_4H_2O_3-H^+$, *m/z*
99), and pthalic anhydride ($C_8H_4O_4-H^+$, *m/z* 149) in aged smoke (Yokelson et al., 2003, 2009; Akagi et al., 2012; Müller
et al., 2016; Bruns et al., 2017; Hartikainen et al., 2018). Calibration factors for these species are calculated from measured
or estimated proton-transfer rate constants. In this study, substantial increases in $C_4H_4O_3-H^+$ (*m/z* 101) and $C_4H_8O_2-H^+$
(*m/z* 89) are also observed. $C_4H_4O_3-H^+$ could be 5-hydroxy-2(5H)-furanone (or simply hydroxy furanone), its tautomer
malealdehydic acid, or succinic anhydride, whereas $C_4H_8O_2-H^+$ is likely a $C_4$ hydroxy carbonyl (see Sections 3.1 and) Liu
et al., 1999; Bierbach et al., 1994, 1995; Alvarez et al., 2009; Aschmann et al., 2011, 2014; Strollo and Ziemann, 2013; Zhao
and Wang, 2017). Due to uncertainties in these assignments, calibration factors for these species are calculated using estimated



proton-transfer rate constants derived from molecular formula relationships (uncertainty to within a factor of 2, Sekimoto et al., 2017).

The I-CIMS utilizes a "soft" chemical ionization source that forms iodide clusters with polarizable analyte molecules (Huey et al., 1995; Lee et al., 2014). The instrument used here was operated in a similar configuration to that described in Krechmer et al. (2016). To generate reagent ions, 2 SLPM of clean $N_2$ from dewar blow-off was run over a methyl iodide permeation tube and ionized using a Polonium-210 ionizer and into an ion molecule reaction region (IMR). The I-CIMS measured gas-phase signals from the mini chamber at 1 Hz time resolution. Smoke was diluted with 4 L min$^{-1}$ of clean, humidified air at a 4:1 ratio to minimize reagent ion depletion. A constant flow of isotopically labeled formic acid was delivered to the instrument to measure consistency of response. Reported here are I-CIMS measurements normalized to $1 \times 10^6$ counts per second of the reagent ion signal at m/z 126.905 (normalized counts per second, ncps). Due to unavailability of standards, I-CIMS data are not reported in mixing ratios. Secondary NMOG measured by I-CIMS are assigned based on modeling results and previous literature.

### 2.3.2 $NO_x$ Measurements

NO, $NO_2$, and HONO were measured by an open-path Fourier transform infrared spectrometer (OP-FTIR) as described by Selimovic et al. (2018). The OP-FTIR was located on the platform and sampled smoke across the diameter of the stack. The OP-FTIR provides fast measurements and avoids potential sampling artifacts due to sample line losses. OP-FTIR measurements were used for most experiments. $NO_2$ and HONO were also measured by the NOAA Airborne Cavity Enhanced Spectrometer (ACES) as described by Zarzana et al. (2018). ACES was located on the platform and sampled smoke through a 1 m Teflon inlet. These data were used when OP-FTIR data were unavailable, or when $NO_x$ emissions were below OP-FTIR detection limits. Supplementary NO measurements were provided by a custom-built chemiluminescence instrument located in a room on the burn chamber floor. A full description of that instrument is provided by (Stockwell et al., 2018).

$NO_x$ was not measured from the mini chamber; consequently, initial $NO_x$ is estimated based on the integrated $NO_x$/acetonitrile ratios measured from the stack during mini chamber filling. Initial HONO mixing ratios are estimated similarly.

### 2.4 Box Model Implementation and Evaluation

NMOG oxidation processes were simulated using the Framework for 0-D Atmospheric Modeling (F0AM v 3.1, https://sites.google.com/site/wolfegm/models, Wolfe et al., 2016). A modified NMOG oxidation mechanism is applied based on the Master Chemical Mechanism (MCM v. 3.3.1, Jenkin et al., 1997, 2003, 2015; Saunders et al., 2003). NMOG chemistry and dilution are assumed to follow the first-order differential equation described by Eqn 1.

$$\frac{dC_i}{dt} = \sum_m^n P_m r_m - \sum_m^n R_m r_m - k_i C_i \qquad (1)$$

Where $C_i$ is the concentration of species $i$, $P_m$ is the stoichiometric coefficient of reaction $m$ leading to the formation of species $i$, $R_m$ is the stoichiometric coefficient of reaction $m$ leading to the loss of species $i$, $r$ is the elementary rate of reaction





$m$, and $k_i$ is a first-order dilution rate constant. A constant dilution term is applied to all species based on the measured loss rate of acetonitrile.

Photolysis rates are calculated using literature cross-sections and quantum yields of relevant organic and inorganic species

(Atkinson et al., 2006; Burkholder et al., 2015). The measured 254 nm photon flux ($3\times10^{15}$ photons cm$^{-2}$ s$^{-1}$) is scaled by a factor of 1.5 in order to reproduce the measured OH-loss of d-butanol. No other changes are applied to the rate constants or reactions of MCM v. 3.3.1. To account for other reactive NMOG, literature mechanisms for furan, 2-methylfuran, 2,5-dimethylfuran, furfural, 5-methylfurfural, and guaiacol are included. A full description of these modifications are provided in Section 3.2.1. Table S1 summarizes the photolysis frequencies and OH, O$_3$, and NO$_3$ rate constants for the NMOG species

modeled here.

The model is evaluated against two fires - F26 (Engelmann spruce duff) and F38 (ponderosa pine litter). These two fires represent different extreme cases in NO$_x$ and NMOG composition and were chosen in order to assess the extent to which the variability of primary NMOG influences secondary NMOG formation. Figure S2 illustrates how F38 and F26 compare to other fires measured in this study. F26 was a unique burn characterized by low-temperature smoldering combustion, which resulted in

low NO$_x$ emissions (NO$_x$/NMOG ~ 0.02 mol/mol) and a NMOG profile with high contributions from oxygenated aromatics such as phenol, cresol, and guaiacol. F38 was representative of many of the burns presented here and was characterized by a mixture of high-temperature flaming combustion and low-temperature smoldering. NO$_x$ emissions were substantially higher than those of F26 (NO$_x$/NMOG ~ 0.32).

The model is initialized with the concentration of 47 primary NMOG species measured by PTR-ToF-MS (Table S1). NO,

NO$_2$, and HONO were initialized based on stack measurements as described in Section 2.3.2. Ozone measurements were conducted using a 2B Technologies ozone monitor (model 202), which exhibited significant interferences upon the addition of smoke or hexafluorobenzene (added as a supplemental dilution tracer). The monitor measures the absorption of O$_3$ at wavelengths that are also absorbed by many primary NMOG. Here, ozone measurements are only used to initialize the model with initial O$_3$ mixing ratios. These initial conditions are based on the signal measured prior to smoke injection and ranged

between 5 - 10 ppb.

Ozone is continuously added to the chamber over the course of an experiment. This input of ozone is included in the model by applying a constant ozone concentration to the dilution term of Eqn. 1. It is estimated that the dilution stream contained ~ 70 ppb of ozone based on the expected output from the ozone generator (~1 ppm) and the measured dilution rate. This input of ozone reproduces the ozone signal measured during dark control experiments to within 20% (Fig. S1).

NO and NO$_2$ are both emitted from fires. Once injected into the chamber, NO will react with O$_3$ to generate NO$_2$. In the atmosphere, NO and NO$_2$ will rapidly cycle owing to NO$_2$ photolysis. In the mini chamber, NO$_2$ photolysis is reduced since NO$_2$ does not strongly absorb at 254 nm (the absorption cross-section at 254 nm is a factor of 64 smaller than the peak absorption at 400 nm, Burkholder et al., 2015). Consequently, radical reactions (e.g., RO$_2$ + NO) and NO$_x$ loss processes (e.g. PAN formation) are likely to be sensitive to the initial NO/NO$_2$ ratio. The initial NO/NO$_2$ ratio is estimated assuming that the

NO$_x$ mixture measured from the stack reacts with a constant mixing ratio of ozone (10 ppb) for 10 minutes (the approximate





mixing periods for F26 and F38). This analysis yields a $NO_x$ distribution that is $\geq$ 95% $NO_2$. This $NO_x$ distribution is applied to the model as an initial condition.

The model output is compared to the temporal profiles of NMOG measured by PTR-ToF-MS and I-CIMS. The mechanism is also tested against aircraft measurements of a small biomass burning plume. Müller et al. (2016) modeled the evolution of

NMOG and formation of ozone from a small agricultural fire during the 2013 DISCOVER-AQ campaign. The model employed chemistry from the MCM v 3.3.1 and a simplified scheme to represent furfural and furan chemistry. Only the decay of furfural was modeled, whereas furan oxidation was assumed to form butenedial. The same model used by Müller et al. (2016) is employed here, except that the mechanism includes the updated reactions of furan, 2-methylfuran, 2,5-dimethylfuran, furfural, 5-methylfurfural, and guaiacol (Section 3.2.1). The initial conditions of phenol and cresol, which are represented in MCM v

3.3.1, are also specified.

## 3    Results and Discussion

### 3.1    Oxidation Product Measurements

Figure S2 summarizes the initial $NO_x$ and NMOG concentrations, $NO_x$/NMOG ratio, and NMOG composition for all the mini chamber experiments sampled by PTR-ToF-MS and I-CIMS. NMOG composition is reported as the fraction of NMOG

signal attributed to high temperature, low temperature, and duff pyrolysis as described by Sekimoto et al. (2018). High temperature pyrolysis results in higher emissions non-functionalized hydrocarbons, such as benzene, while low temperature pyrolysis results in higher emissions of oxygenated species, such as methoxy phenols. Duff pyrolysis generates a NMOG distribution that is similar to the distribution from low temperature pyrolysis, except that the unique composition of duff results in higher emissions of nitrogen-containing compounds.

The initial conditions for mini chamber experiments varied drastically depending on chamber dilution, fuel type, and burn conditions (Fig. S2). The $NO_x$/NMOG ratio varied over several orders of magnitude (0.01 – 1.2) with NMOG loadings ranging from 90 – 900 ppb. Fires produced varying distributions of NMOG owing to the extent of high and low temperature pyrolysis, while $NO_x$ concentrations varied depending on fuel nitrogen content (Burling et al., 2010) and the extent of flaming combustion (Sekimoto et al., 2018). For the remaining discussion, only experiments with NMOG loadings < 300 ppb (12 total) are analyzed

in order to evaluate biomass burning chemistry at the lowest calculated primary OH reactivities (OHR), since high OHR values can lead to deviations between chamber and ambient chemistry (Peng et al., 2016, 2019). The experiments reported here have estimated OHR between 18 - 70 $s^{-1}$.

Figure 1 shows the temporal evolution of select NMOG measured by PTR-ToF-MS and I-CIMS during F38 (Ponderosa Pine). The left column shows the decay of primary NMOG, while other columns show the temporal profiles of secondary

NMOG exhibiting significant enhancements in PTR-ToF-MS and I-CIMS spectra. Secondary NMOG are classified based on temporal profile - those exhibiting a relatively fast increase in signal are classified as "fast-forming" products, while those formed more gradually over the course of an experiment are classified as "slow-forming" products. These designations are chosen based on the oxidation time scales that are likely to be observed in ambient biomass burning plumes. Here, OH expo-





sures are estimated by the decay of d-butanol, then converted to atmospheric-equivalent timescales assuming and atmospheric

OH concentration of $1.5 \times 10^6$ molecules cm$^{-3}$. Fast-forming products generally peaked within 10-20 hrs of atmospheric-equivalent oxidation, while slow-forming products exhibited no maxima.

The loss of reactive primary species occurs quickly (< 20 hrs of atmospheric-equivalent oxidation). Most species (e.g. dimethylfuran and guaiacol) follow an exponential decay consistent with radical loss pathways; however, some species, such as furfural and 5-methylfurfural, show faster decay immediately following the initiation of lights. The lights employed in these

experiments (Ultra-Violet Products, Inc.) emit a narrow band at 254 nm, which is capable of photolyzing furanaldehydes and other absorbing species (Hiraoka and Srinivasan, 1968; Gandini et al., 1976). Based on the measured flux, literature cross section, and quantum yield of furfural ($\sim 0.05$, Hiraoka and Srinivasan, 1968; Gandini et al., 1976), the furfural photolysis rate constant is estimated to be 0.13 s$^{-1}$, which is orders of magnitude higher than the photolysis frequency expected under ambient, summer-time conditions ($7.2 \times 10^{-4}$ s$^{-1}$, Colmenar et al., 2015). Furfural photolysis leads to the formation of other

highly reactive furans (e.g. furan and 2-methylfuran) at modest yield (<0.3, Gandini et al., 1976)), and these products may contribute to the OH reactivity early on during each experiment.

The majority of secondary NMOG observed by the PTR-ToF-MS are formed slowly as "slow-forming" products. Previous studies employing PTR-ToF-MS have identified acetic acid, formic acid, and maleic anhydride as major products (Müller et al., 2016; Bruns et al., 2017; Hartikainen et al., 2018). Formic and acetic acid are primary species, but also form from OH

oxidation of alkene and aromatic species (e.g., Jenkin et al., 1997, 2003, 2015; Saunders et al., 2003; Millet et al., 2015). Maleic anhydride is known to form from the oxidation of aromatics, but it has also been observed to form from the oxidation of furans (Bierbach et al., 1995). The formation of slow-forming products occurs over timescales > 20 hrs, which is significantly longer than the evolution of fast-forming (e.g. $C_5H_6O_3$, $\sim 4$ hrs) and losses of the most reactive primary NMOG (e.g. 2-methylfuran and guaiacol, $\sim 4$-10 hrs). This may be an indication of multi-generational oxidation, or early-generation formation via slowly-

reacting primary species.

Figure 2 shows the resulting changes in NMOG composition measured by the PTR-ToF-MS after $\sim 12$ hrs of atmospheric-equivalent oxidation. Data are shown in terms of primary carbon changes ($\Delta_C$NMOG, %), which is the fraction of initial NMOG carbon measured by the PTR-ToF-MS that was consumed (reactant) or formed (product) for a given species or group of species. "Reactant" and "product" classifications were determined based on whether a detected mass exhibited an increase

or decrease in dilution-corrected signal after 12 hours of atmospheric-equivalent oxidation. The bars in panel B represent species-aggregate measurements for each of the 12 fires highlighted in Fig. S2. Panel A shows an average of all secondary NMOG formed after 12 hrs of atmospheric-equivalent oxidation. Across most experiments, furans and oxygenated aromatics were the primary NMOG with the greatest carbon losses. Decreases in oxygenated aromatics were mostly driven by losses of guaiacol, methyl guaiacol, and catechol. Losses of furans were primarily due to decreases in furan, 2-methylfuran, 2,5-

dimethylfuran, furfural, and 5-methylfurfural. These observations are consistent with previous PTR-ToF-MS measurements of aged wood burning smoke (Bruns et al., 2017; Hartikainen et al., 2018). The contribution from biogenic species in this study was variable, and largely dependent on composition (Hatch et al., 2015, 2017) and the extent to which monoterpenes were emitted from the distillation phase of combustion (Koss et al., 2018; Sekimoto et al., 2018). The detected masses associated





with the alkene category exhibited both apparent formation and consumption across various experiments. Alkenes are expected

to react quickly with OH; thus, observed increases likely point to a species misassignment. It is likely that this increase in signal is the result of PTR-ToF-MS fragmentation of oxygenated NMOG. For example, some alcohols, acids, and certain aldehydes may fragment in the PTR-ToF-MS to masses that are typically associated with alkenes (e.g. *m/z* 69, Buhr et al., 2002; Pagonis et al., 2019). These species are likely formed from the oxidation of biomass burning precursors and potentially interfere with alkene detection.

Across most experiments, small oxygenates and acids (C<5) were the predominant secondary NMOG detected by PTR-ToF-MS (Fig. 2B). As mentioned previously, formic acid, acetic acid, and maleic anhydride are several species with the largest relative increase at longer oxidation time scales (Fig. 2A); however, formaldehyde, acetaldehyde, acetone, and several $C_4O_xH_y$ species increased significantly. The formation of $C_4O_xH_y$ species follows a similar temporal pattern as maleic anhydride, which may point to similarities in species functionality or formation pathways.

In contrast to the PTR-ToF-MS measurements, the I-CIMS measured a mix of fast- and slow-forming products (Fig. 1). The I-CIMS measured a small fraction of the primary NMOG, so only secondary NMOG are discussed here. Fast-forming products included pyruvic acid ($C_3H_4O_3$), $C_xO_3H_y$ compounds, and masses likely corresponding to nitroaromatics. Previous studies have reported the formation of nitroaromatics from OH oxidation of catechol and guaiacol in the presence of $NO_2$ (Lauraguais et al., 2014; Finewax et al., 2018). $C_7H_7NO_4$ is consistent with the formation of nitroguaiacol, which has been observed in

the gas and particle phase from guaiacol + OH (+$NO_2$) chemistry (Lauraguais et al., 2014). Finewax et al. (2018) studied the OH oxidation of catechol and observed the formation of nitrocatechol with a molar yield of 0.3. Very little nitrocatechol was observed by the PTR-ToF-MS and I-CIMS, which is likely due to the high aerosol loadings in the chamber ($> 50 \ \mu g \ m^{-3}$) and high affinity of nitrocatechol to partition to aerosols and surfaces (Finewax et al., 2018).

Figure 3 summarizes the changes in I-CIMS product spectra relative to the total signal measured prior to photochemistry.

Shown are the product distributions for short (4 hr), medium (12 hr), and longer (24 hr) atmospheric time scales. Because not all masses could be calibrated, normalized difference spectra (oxidized minus primary) are presented in order to illustrate the ions with the largest relative increases. In general, the secondary NMOG measured by I-CIMS tend to be multi-functional. The smallest observed oxygenate is formic acid ($CH_2O_2$), while larger molecules tend to be $C_{2-5}H_xO_3$ and $C_{4-5}H_xO_4$ compounds. After 4 hrs of oxidation, the largest enhancements are due to fast-forming products such as $C_3H_4O_3$, $C_4H_4O_3$,

and $C_5H_6O_3$. The relative importance of these masses decreases at longer time scales, whereas the relative abundance of smaller oxygenated increases (e.g. $CH_2O_2$ and $C_2H_4O_3$).

The signal intensity of the fast-forming products (specifically $C_4H_4O_3$ and $C_5H_6O_3$) suggests that these species result from the oxidation of abundant, fast-reacting NMOG precursors with carbon number $\geq$ 4. Modeling results presented in Section 3.2 support the assignment of these products as hydroxy furanone ($C_4H_4O_3$) and methyl hydroxy furanone ($C_5H_6O_3$). Based

on the loss of primary NMOG (Fig. 2), it is likely that these species are formed from the oxidation of furans, oxygenated aromatics, or other fast reacting NMOG with C≥4. Several studies have investigated the oxidation of furan species and shown that hydroxy furanone + tautomer ($C_4H_4O_3$) and methyl hydroxy furanone + tautomer ($C_5H_6O_3$) are major products formed from the oxidation of furfural, furan, 2-methylfuran, and 2,5-dimethylfuran (e.g. Bierbach et al., 1994, 1995; Alvarez et al.,





2009; Aschmann et al., 2011, 2014; Strollo and Ziemann, 2013; Zhao and Wang, 2017). Secondary NMOG measured from the
OH chemistry of oxygenated aromatic species are largely carbon-retaining (C ≥ 6), though $C_4H_4O_4$ ring fragments have been
measured in low-$NO_2$ catechol oxidation (Yee et al., 2013).

Notably, inspection of Figures 2A and 3 shows that there is little overlap in the species measured by PTR-ToF-MS and I-
CIMS. The only masses that exhibit significant enhancements in both spectra are formic acid ($CH_2O_2$) and $C_4H_4O_3$. Moreover,
there are significant differences in the temporal evolution of $C_4H_4O_3$. As discussed above, $C_4H_4O_3$ measured by I-CIMS most
likely corresponds to hydroxy furanone, a fast-forming product (Fig. 1). In contrast, $C_4H_4O_3$ as measured by the PTR-ToF-
MS exhibits a temporal profile resembling that of a slow-forming product. It is likely that this species is succinic anhydride,
which is structurally similar to maleic anhydride and could be formed from multi-generation chemistry. The differences in
these profiles suggests that the there are at least two different $C_4H_4O_3$ species present, that PTR-ToF-MS and I-ToF-MS are
sensitive to different biomass burning oxidation products, and that both instruments are needed in order to measure important
secondary NMOG.

Previous studies and the mass spectra in Figures 2 and 3 show that furan chemistry plays a significant role in the atmospheric
chemistry of biomass burning plumes (Bruns et al., 2017; Hartikainen et al., 2018; Gilman et al., 2015; Hatch et al., 2015). In
Section 3.2, furan chemistry is incorporated into a 0-D box model to help interpret the observed small-chamber measurements
and previously measured ambient biomass burning plumes.

## 3.2 NMOG Box Modeling

### 3.2.1 Mechanism Incorporation

The box model described in Section 2.4 employs NMOG chemistry based on the Master Chemical Mechanism (MCM v.
3.3.1, Jenkin et al., 1997, 2003, 2015; Saunders et al., 2003). The MCM explicitly represents the chemistry of biogenic,
alkyl, aromatic, and oxygenated aromatic species. The laboratory measurements described in Section 3.1 demonstrate that
heterocyclic hydrocarbons, such as the furans, could significantly contribute to secondary NMOG formation. The following
discussion motivates and describes mechanism development aimed at expanding the MCM representation of biomass burning
OH chemistry.

Figure 4 shows the breakdown in OH reactivity of the primary NMOG measured by PTR-ToF-MS during the Firelab in-
tensive (Koss et al., 2018). Each bar represents a fraction of the total calculated OH reactivity, as represented by Equation
370  2.

$$f_{OHR,i} = \frac{k_i \times C_i}{\sum_i^n k_i \times C_i} \qquad (2)$$

Where $i$ is the species of interest, $n$ is the number of species measured by the PTR-ToF-MS, $k$ is the OH rate constant,
and $C$ is the average concentration measured during a burn. Shown are median, average, $25^{th}$, and $75^{th}$ percentiles for all
the burns reported by Koss et al. (2018). The color of each bar indicates if a compound is included or missing from the





MCM. Species identifications, and the isomer contributions to each detected mass, were determined by Koss et al. (2018)
for four fuel types (Douglas fir, Engelmann spruce duff, subalpine fir, and sage) using gas-chromatography pre-separation
(GC-PTR-ToF-MS). Additional evidence for species identification was provided by other measurement techniques (e.g., I-
CIMS, gas-chromatography electron-impact mass spectrometry, and OP-FTIR). On average, the isomer contribution to a given
mass measured by PTR-ToF-MS varied by only 11%. To be consistent with Koss et al. (2018), it is assumed that the NMOG

contribution to each mass detected by PTR-ToF-MS follows the average distribution measured by GC-PTR-ToF-MS.

For most NMOG measured by PTR-ToF-MS, the contribution to total primary OH reactivity varied by only 25%. Notably,
the contribution from the sum of monoterpenes varied by a factor $> 2$. Monoterpenes, as well as isoprene and sesquiterpenes,
were primarily emitted at the beginning of an experiment, prior to combustion, due to distillation processes associated with
fuel heating (Sekimoto et al., 2018). This "distillation phase" was most pronounced in fires containing greater amounts of

canopy material, or fuel types known to be strong monoterpene emitters (e.g. pines). Other NMOG were primarily emitted
due to pyrolysis processes. For example, Sekimoto et al. (2018) found that the proportions of NMOG emitted during low-
and high-temperature pyrolysis did not strongly vary by fuel type. In ambient fires, the contribution of monoterpenes to the
total OH reactivity will likely differ from the contributions reported here, owing to the different burning process by which
monoterpenes and other NMOG are emitted. In this study, the primary monoterpene isomers measured by GC-PTR-ToF-MS

for pine fuel types were camphene, $\alpha$-pinene, $\beta$-pinene, and limonene followed by smaller amounts of tricyclene and $\alpha$-
terpinene (Fig. S3). No other NMOG detected by the GC-PTR-ToF-MS produced signals at *m/z* 137, which is the primary
ion used to quantify monoterpene emissions (Fig. S3 and Koss et al., 2018). The MCM represents the chemistry $\alpha$-pinene,
$\beta$-pinene, and limonene, but does not explicitly describe other important monoterpenes (e.g. camphene). For simplicity, the
model described subsequently represents the chemistry of the sum of monoterpenes through the $\alpha$-pinene mechanism.

On average, the MCM v 3.3.1 captures only $\sim 60\%$ of the primary OH reactivity measured by the PTR-ToF-MS. The MCM
generally lacks information about furan species and substituted aromatics, such as guaiacol and methyl guaiacol. Previous work
has shown that furans constitute a significant fraction of the total primary OH reactivity (e.g. Gilman et al., 2015; Hatch et al.,
2015, 2017; Koss et al., 2018); however, no studies have included the known mechanisms of these species when modeling
biomass burning smoke chemistry. 5-methylfurfural and 2-dimethylfuran are the two largest contributors to this "missing

reactivity" and account for nearly 10% of the calculated total reactivity. Up to 75% of the calculated primary OH reactivity
can be accounted for by including the chemistry of furan, 2-methylfuran, 2,5-dimethylfuran, furfural, 5-methylfurfural, and
guaiacol, along with the known species represented in the MCM. Much of the remaining reactivity is tied into species whose
chemistry has not been extensively studied, including lesser abundant furans and oxygenated aromatics.

The mechanisms of select furan and oxygenated species are incorporated to the MCM based on previous work summarized in

Figures S4 - S9 (Bierbach et al., 1995; Alvarez et al., 2009; Aschmann et al., 2011, 2014; Strollo and Ziemann, 2013; Zhao and
Wang, 2017). In total, 65 reactions are added. The products resulting from the OH oxidation of furan and 2-methylfuran were
first investigated by Bierbach et al. (1995) and later generalized to 3-methylfuran, 2,3-dimethylfuran, and 2,5-dimethylfuran
by Aschmann et al. (2014) and Strollo and Ziemann (2013). Figure 5 shows the generalized furan oxidation scheme. Furan
oxidation is initiated by an OH addition to the 2, 5, or 3-position. Addition to the 2 or 5-position is most favorable and results





in pathways where substituents are either retained or lost (henceforth referred to as the loss and retention pathways). The retention pathway (path A in Fig. 5) leads to reactive unsaturated 1,4-dicarbonyls (e.g. 1,4-butenedial from furan oxidation), whereas the loss pathway (paths B1 and B2 in Fig. 5) results in the formation of hydroxy furanones and unsaturated carbonyl-acids (Aschmann et al., 2014; Strollo and Ziemann, 2013). The loss pathway becomes more dominant with higher number of substituted methyl groups.This study assumes branching ratios of [0.7 (A), 0.3 (B1)] for furan, [0.31 (A), 0.39 (B1), 0.31 (B2)] for 2-methylfuran, and [0.27 (A), 0.73 (B1)] for 2,5-dimethylfuran (Aschmann et al., 2014).

$RO_2$ reactions leading to the formation of carbonyls are implemented based on the mechanisms proposed by Aschmann et al. (2014) and Bierbach et al. (1994) (Figures S4 - S6). It is assumed that $RO_2$ species undergo reactions with $HO_2$, NO, and other $RO_2$ radicals. Other pathways, such as $RO_2$ + $NO_2$ and $RO_2$ + $NO_3$, are not included; however, these reactions could be important for acyl $RO_2$ species (Orlando and Tyndall, 2012; Peng et al., 2019). For $RO_2$ + NO reactions, it is assumed that the alkoxy radical quickly decays (either by thermal degradation or reaction with $O_2$) to form carbonyls. Aschmann et al. (2014) did not report species consistent with alkoxy isomerization; thus, these reactions are ignored. Similarly, $RO_2$ + $RO_2$ reactions are assumed to only form alkoxy radicals, which may subsequently degrade to from carbonyls. Peroxides are assumed to be the only products of $RO_2$ + $HO_2$ reactions. These species are assumed to undergo photolysis to also form carbonyls. The generic MCM rate constants are applied for $RO_2$ + $HO_2$ and $RO_2$ + NO reactions ($k_{RO2NO}$ = 2.7 $\times 10^{-12}$ exp(360/T) cm$^3$ molec $^{-1}$ s$^{-1}$, $k_{RO2HO2}$ = 2.91 $\times 10^{-13}$ exp(1300/T) cm$^3$ molec $^{-1}$ s$^{-1}$). The assumed rate constants for $RO_2$ + $RO_2$ reactions are chosen based on the reactions of structurally similar $RO_2$ radicals reported in the MCM. $RO_2$ H-shift isomerization (autoxidation) is not represented in MCM v. 3.3.1, or in the reactions described here. $RO_2$ isomerization becomes competitive when the bimolecular lifetime of $RO_2$ is on the order of 10s (Crounse et al., 2013; Praske et al., 2018). The bimolecular lifetime of $RO_2$ radicals in the chamber is estimated to be $\sim$ 10s; consequently, $RO_2$ isomerization could play a role for certain species.

Following $RO_2$ reaction, it is assumed that the second-generation products continue through the chemistry prescribed by MCM v 3.3.1. Unsaturated dicarbonyls, such as 1,4-butenedial, and the tautomers of hydroxy furanones are represented in the MCM (Figures S4 - S6). Here, it is assumed that hydroxy furanones undergo the same reactions as the corresponding tautomer, which ultimately leads to anhydride formation. Maleic anhydride is a multi-generational product in the OH oxidation of furan (Bierbach et al., 1995) and is considered to be a significant product of hydroxy furanone oxidation (Bierbach et al., 1994).

No experimental studies have evaluated the OH oxidation mechanism of furfural or 5-methylfurfural. Zhao and Wang (2017) found via theoretical quantum chemistry calculations that OH likely adds to the 2 or 5-position or abstracts a hydrogen from the aldehyde. The resulting reactions follow loss and retention pathways similar to the general mechanism for methyl-substituted furans (Fig. 6). When OH adds to the 2-position, the ring most likely opens to form an unsaturated tri-carbonyl (retention, path A). When OH adds to the 5-position, the resulting peroxy radical may react with $HO_2$, NO, or other other $RO_2$ species to form a hydroxy furanone / carbonyl acid mixture (loss, path B). Hydrogen abstraction from the aldehyde group is believed to ultimately result in the formation of maleic anhdyride (loss, path C). Zhao and Wang (2017) estimate furfural + OH branching ratios of 0.37 for channel A, 0.6 for channel B, and 0.03 for channel C. The same branching ratios are applied here, but a discussion of secondary NMOG sensitivity to the assumed furfural mechanism is provided in the supplemental information.





The 5-methylfurfural + OH mechanism has not been studied and is assumed to have branching ratios similar to furfural. $RO_2$ reactions are implemented based on the mechanisms proposed by Zhao and Wang (2017) (see Figures S7 - S9).

Furfural strongly absorbs at 185 and 254 nm (Gandini et al., 1976; Hiraoka and Srinivasan, 1968; Ferreira da Silva et al., 2015). Smaller amounts may be lost by photolysis at wavelengths > 300 nm (Colmenar et al., 2015). Approximately 15% of furfural photolysis leads to the formation of furan and CO, while the remaining results in the formation of propyne, CO, and

other $C_3$ compounds (Gandini et al., 1976). The photolysis rate constant is calculated based on the cross sections reported by Ferreira da Silva et al. (2015) and quantum yield reported by Gandini et al. (1976) ($\sim$ 0.6).

Few studies have evaluated the OH oxidation mechanism of guaiacol. Yee et al. (2013) identified products from low-$NO_x$ guaiacol oxidation, but did not calculate product yields. Lauraguais et al. (2014) studied guiacol oxidation in the presence of $NO_x$ and observed substantial SOA formation. The only reported gas-phase species were a suite of nitroguaiacols composed

primarily of 3- and 6-nitroguaiacol (6% yield) and 4-nitroguaiacol (10% yield). The mechanism proposed by Lauraguais et al. (2014) is applied here assuming a 16% yield of nitroguaiacol species (Fig. S10).

The OH loss of d-butanol is also included in the model to validate OH concentrations in the chamber. The OH oxidation of d-butanol is assumed to form a single, non-reactive species.

### 3.2.2 Model/Measurement Evaluation

Figure 7 shows the model comparison with PTR-ToF-MS measurements of d-butanol and select primary NMOG for F38. Overall, there is good agreement between the measurements and model output for most NMOG. The excellent agreement between the measured and modeled loss of d-butanol demonstrates that OH concentrations in the chamber are well-represented by the model. Similar agreement is observed for F26, which had an initial $NO_x$/NMOG ratio that was an order of magnitude lower than that of F38 (Figure S11). Figure 8 shows the model output compared to the observed profiles of secondary NMOG

measured during F38. The equivalent for F26 is presented in Figure S12. Panel A shows measurements of $C_4H_2O_3$ (maleic anhydride) as measured by PTR-ToF-MS, whereas panels B and C show I-CIMS measurements of $C_5H_6O_3$ and $C_4H_4O_3$, respectively. The measured secondary NMOG are compared to model outputs of total $C_4H_2O_3$ (i.e., the sum of all species with molecular formula $C_4H_2O_3$), total $C_5H_6O_3$, total $C_4H_4O_3$, and individual NMOG. Figure 8 also show model runs with the initial conditions of furan, 2-methylfuran, 2,5-dimethylfuran, furfural, 5-methylfurfural, and furanone set to zero. The PTR-

ToF-MS measurements of $C_4H_2O_3$ are reported in units of ppb and can therefore be quantitatively compared to model output. I-CIMS measurements are reported as ncps; consequently, only qualitative comparisons are drawn based on similarities in model/measurement temporal profiles.

$C_4H_2O_3$, $C_5H_6O_3$, and $C_4H_4O_3$ were shown in Section 3.1 to constitute some of the most abundant photochemical products observed by PTR-ToF-MS and I-CIMS. The model runs in Figures 8 and S12 demonstrate that furan chemistry significantly

contributes to the modeled formation of these secondary NMOG. The model also supports the inference that these masses correspond to measurements of maleic anhydride, methyl hydroxy furanone, and hydroxy furanone, respectively. This is most evident by comparing the shape of the temporal profiles between the measurements and modeled output. Maleic anhydride is the only species in the MCM with chemical formula $C_4H_4O_3$, and the model generally captures the peak $C_4H_4O_3$ signal





after $\sim 20$ min of oxidation to within the uncertainty of the measurement ($\sim 50\%$). Several species with chemical formula
$C_5H_6O_3$ are represented in the MCM. Here, the modeling output is dominated by methyl hydroxy furanone and its tautomer,
$\beta$-acetylacrylic acid. Finally, I-CIMS measurements of $C_4H_4O_3$ are best captured by the model output of hydroxy furanone
and its tautomer, malealdehydic acid.

The temporal profile of $C_7H_7NO_4$ is also well described by model output of nitroguaiacol (Fig. S13). Nitroaromatics are
formed by the reaction of $NO_2$ with the o-semiquinone radical resulting from OH abstraction of the phenolic hydrogen. $NO_2$
was abundant at the beginning of mini chamber experiments performed on flaming emissions (Fig. S2); thus, nitroaromatics
are expected to be present in many burns studied here. Note that only the results from F38 are shown. Very little formation of
nitroguaiacol was observed in F26, owing to the relatively low amount of $NO_x$ emitted from the smoldering combustion of
Engelmann spruce duff.

The model provides insights into the formation pathways of important secondary NMOG, which could be used to place
constraints on plume properties. For example, measurements of maleic anhydride, hydroxy furanone, and methyl hydroxy
furanone could be used as proxies to estimate plume age since the furanones will likely be enhanced in younger plumes,
whereas maleic anhydride will likely be enhanced in aged plumes. In plumes containing high proportions of furans, it could be
feasible to evaluate furan chemistry to derive important modeling constraints, such as OH exposures.

Despite the general success of the model in reproducing observations, several differences exist as described below. First,
the model output exhibits a faster decay of furans than what is observed by PTR-ToF-MS (Figure 7). This likely reflects the
uncertainty associated with the mass assignment of furan species. Using GC-PTR-ToF-MS, Koss et al. (2018) showed that
nearly 50% of the primary signal at $C_5H_6O-H^+$ (mass of 2-methylfuran) and $C_6H_8O-H^+$ (mass of 2,5-dimethylfuran) is
associated with unidentified oxygenates. These unidentified species likely have different reactivities towards OH; thus, the
temporal profile of these masses are unlikely to match with the model output.

The decay of furfural is distinct from other furans because several processes contribute to temporal profile of $C_5H_4O_4-H^+$.
Notable, there appears to be a fast decay of $C_5H_4O_4-H^+$, followed by a slower decay after 10s of oxidation. The model
generally captures the fast decay of $C_5H_4O_4-H^+$, which is almost entirely due to photolysis of furfural. This degree of
photolysis is different from the real atmosphere and results from furfural's exceptionally large cross section at 254 nm ($>$
$5\times10^{-17}$ cm$^2$, Ferreira da Silva et al., 2015). The slower decay appears to result from an interference of another NMOG. This
is supported by the I-CIMS which measured the formation of a mass with formula $C_5H_4O_4-I^-$. The I-CIMS is not sensitive to
primary furan species, therefore this species is likely to be a secondary NMOG that is isobaric with furfural. This may explain
why PTR-ToF-MS measurements of $C_5H_4O_4-H^+$ do not quickly decay to zero, as suggested by the model. It is notable that
the formation of this secondary species is significant ($\sim 50\%$ of the signal of primary furfural), which indicates that this is
likely a secondary product formed from an abundant primary NMOG.

Despite the complications imposed by furfural photolysis, other furans and oxygenated aromatics do not exhibit strong
absorption and are expected to be lost mostly by reaction with OH. Other absorbing species, such as methyl ethyl ketone and
benzaldehyde, exhibit modeled photolysis losses on the order of 30%, which is likely a more typical fraction for other photo-





active species. More details comparing the chamber results to chemistry of ambient biomass burning plumes are provided in Section 3.2.3.

At the beginning of each experiment, PTR-ToF-MS measurements show a sharp increase in $C_4H_2O_3$ that is not readily captured by the model (Figure 8). This increase could result from fast formation of maleic anhydride, or is possibly another species with molecular formula $C_4H_2O_3$. The model underpredicts maleic anhydride concentrations towards the end of the experiment, which likely points to additional sources of maleic anhydride that are not included in the model. The model reproduces peak maleic anhydride concentrations in F38 (Fig. 8A), but over predicts peak maleic anhydride concentrations by

a factor of 1.6 in F26 (Fig. S12A).

    After $\sim 30$ min of oxidation, the model sum of $C_4H_4O_3$ approaches zero, yet the I-CIMS signal remains elevated (Figures 8C). This may indicate that a slow-forming product is detected by I-CIMS, or that the OH rate constant of hydroxy furanone is overestimated. As discussed in Section 3.1, the PTR-ToF-MS detects a slow-forming product that likely corresponds to succinic anhydride (Fig. 1).The I-CIMS is sensitive to anhydrides; thus, it is possible that the elevated signal at longer oxida-

tion timescales corresponds to the measurement of succinic anhydride. Succinic anhydride is not represented in the updated mechanism; consequently, no model output is available for comparison.

    Figure 2 shows that small oxygenates are also abundant secondary NMOG (i.e. acetaldehyde and formaldehyde); however, these species are underpredicted by the model by a factor of 10 or more (Fig. S13). This likely reflects additional sources or chemical pathways that are unaccounted for within MCM v 3.3.1. This is complicated by the hundreds of pathways within

the MCM that lead to the formation of small oxygenates, which makes it challenging to isolate key precursors. Furthermore, heterogeneous reactions and other chemical processes may contribute to secondary NMOG formation. Additional work is needed to better describe small oxygenate formation.

    It is noted that without calibrated I-CIMS data, it is difficult to assess whether the budget of these secondary NMOG is fully represented by the model. Although it is expected that furans will be a primary precursor of $C_4H_4O_3$ and $C_5H_6O_3$ in

real biomass burning plumes, the model indicates that other highly reactive species may also contribute to these masses. In F38, furans account for $\sim 80\%$ of the modeled production of $C_4H_4O_3$ and $\sim 90\%$ of the modeled production of $C_5H_6O_3$. In contrast, furans account for $\sim 60\%$ of the modeled production of $C_5H_6O_3$ and $\sim 85\%$ of the modeled production of $C_4H_4O_3$ during F26. The remaining production in the model is attributed to the OH oxidation of oxygenated aromatics - specifically, phenol and cresol. These oxygenated aromatics are more abundant during F26 due to the higher degree of smoldering combus-

tion. These differences highlight the variability of secondary NMOG production and also imply that there could be remaining precursors of $C_4H_4O_3$ and $C_5H_6O_3$. These precursors would have to be highly reactive molecules with carbon number $\geq 4$. There are a number of furans and oxygenated aromatics that could possibly contribute to the formation of these secondary NMOG (e.g. hydroxymethylfurfural, Fig. 4), but whose chemical mechanism remains unknown.

### 3.2.3   Comparison of Chamber Chemistry to Atmospheric Conditions

Section 3.2.2 demonstrates that the high OH environment in the mini chamber can be well-represented by a simple box model. The oxidation processes in the mini chamber are similar to those produced in oxidation flow reactors. Peng et al. (2016,



2019) showed that oxidation flow reactors can be operated under conditions that approximate the chemistry of the atmosphere; however, the use of 254 nm light can lead to photolysis processes that are unlikely to occur under in ambient environments. For example, furfural photolysis is unlikely to play a significant role in the chemistry of real smoke, and $NO_x$ cycling is faster

under ambient photolysis. Furthermore, high OH environments may lead to $RO_2$ fates that differ from what is expected in the ambient. In the atmosphere, the predominant fate of $RO_2$ is reaction with NO or $HO_2$. In wildfires, $RO_2$ reactions with $NO_2$ are also important in forming PAN and other peroxy nitrates (Alvarado et al., 2010). In low $NO_x$ environments, $RO_2 + RO_2$ and $RO_2$ isomerization may also play a role (Crounse et al., 2013; Praske et al., 2018). $RO_2$ isomerization was not included in the mechanism described here. The following discussion compares the modeled chemistry in the chamber to that expected for

an ambient biomass burning plume.

In the atmosphere, primary NMOG are mostly consumed via photolysis or reaction with OH during the daytime, and $O_3$ or $NO_3$ at night. Table S2 shows the estimated contribution of each process to the primary NMOG consumed for F26 and F38. Also shown are the NMOG losses calculated from simulations of the ambient biomass burning plume described by Müller et al. (2016) (see Section 3.2.4 for details of this modeling). For most species, the predominant loss pathway is reaction

with OH. Ozonolysis is negligible for most species, except for a small fraction of $\alpha$-pinene ($\sim$ 1%). On the other hand, significant losses of NMOG occur by photolysis, and to a lesser extent, reaction by $NO_3$. Aside from furfural, loss by photolysis was dominant for acetone and 2,3-butanedione ($>$ 50%), significant for methyl ethyl ketone and benzaldehyde ($\sim$30%), and moderate for hydroxyacetone, glyoxal, methyl glyoxal, formaldehyde, and acetaldehyde ($<$ 10%). In general, photolysis losses were dependent on the relative ratio between the OH rate constant and photolysis frequency at 254 nm (Table S1). Photolysis

losses were greatest for conjugated aldehydes and species with low OH rate constants and high absorption cross-sections. (e.g., acetone, Table S1). Conjugated aldehydes such as furfural are highly reactive towards OH; consequently, losses due to photolysis are notable since these processes represent unintended sinks of potentially important SOA and ozone precursors. In contrast, photolysis losses of other species, such as acetone, is likely less important since these species are less reactive towards OH. It is noted that other species reported here are likely to photolyze, but whose absorption cross-sections at 254 nm have not

been measured (e.g. 5-methylfurfural, Colmenar et al., 2015).

Because numerous species absorb across a wide wavelength spectrum (e.g. acetaldehyde and hydroxyacetone Burkholder et al., 2015), there is some agreement between the photolysis losses estimated in the mini chamber with those expected under ambient conditions. For hydroxyacetone, 2,3-butanedione, and acetaldehyde, reaction by photolysis was comparable to what was estimated for the ambient biomass burning plume described by Müller et al. (2016). In contrast, conjugated aldehydes,

such as furfural and benzaldehyde, are characterized with absorption cross sections that favor shorter wavelengths; consequently, photolysis losses in the mini chamber greatly exceed those expected in ambient plumes. These results highlight the challenges associated with studying multi-day oxidation of biomass burning smoke in environmental chambers. Biomass burning emissions contain a myriad of functionalized NMOG that readily photolyze at wavelengths required to generate high OH environments. Experimental setups employing UVC lights must weigh the option between operating at high OH exposures

(and thus progressing through chemistry quickly) with operating at gradual OH exposures that allow for longer sampling, but higher exposures to UVC light. Similar considerations are made for oxidation flow reactors, although nearly all experiments





are conducted at high OH exposures since sampling is conducted at pseudo steady-state (Peng et al., 2016). To avoid high photolysis exposures, future experiments employing 254 nm light may consider operating with higher ozone concentrations to increase the losses due to reaction by OH. Alternatively, chamber experiments operated at high relative humidity may employ

185 nm light to generate high OH environments (Peng et al., 2016). In both cases, the chemistry will progress quickly, which may be undesireable for chamber experiments using low time-resolution instrumentation. Another approach may be to use UVB or UVA lights and photolyze HONO to generate OH. This approach will reduce photolysis exposures, but may only access 1-2 days of atmospheric-equivalent oxidation.

Unlike primary emissions which are dependent on the balance between photolysis and oxidant concentrations, the formation

of secondary NMOG largely depends on the fate of the $RO_2$ radical. Figure 9 A and B shows a breakdown of the modeled $RO_2$ pathways that contributed to the chemistry of the mini chamber. The bars quantify the total $RO_2$ carbon reacted for the sum of radicals classified by carbon number. The color of each bar represents the fraction of each category that reacted by $RO_2$ + NO, $RO_2$ + $HO_2$, $RO_2$ + $NO_2$, and $RO_2$ + $RO_2$ pathways. Panel C shows the breakdown of $RO_2$ pathways for simulations of the ambient biomass burning plume described by Müller et al. (2016).

In general, the amount of carbon attributed to each $RO_2$ category varied strongly between F26 and F38. As described in Section 2.4, Engelmann spruce duff was burned during F26 and resulted in greater emissions of high carbon (C $\geq$ 6) NMOG, such as phenol, cresol, and guaiacol. Consequently, the distribution of reacted $RO_2$ carbon favors species with higher carbon number. In contrast, ponderosa pine was burned during F38, which resulted in higher emissions of small oxygenates, furans, and non-functionalized aromatics. As a result, a greater fraction of $RO_2$ carbon is associated with lower-carbon species. For

comparison, the ambient biomass burning plume reported by Müller et al. (2016) was characterized by an $RO_2$ distribution that generally decreased with carbon number. Furthermore, the total $RO_2$ carbon that was reacted in the ambient plume ($\sim$ 40 ppb) was lower than that of F26 ($\sim$ 233 ppb) and F38 ($\sim$ 175 ppb).

The $RO_2$ carbon reacted in F38 largely followed two pathways - $RO_2$ + $HO_2$ and $RO_2$ + $NO_2$. The $RO_2$ + $NO_2$ pathway primarily influenced the fate of smaller $RO_2$ radicals (C$\leq$3), leading to the formation of peroxy nitrates. For larger $RO_2$

radicals (C$\geq$4), the dominant pathway was $RO_2$ + $HO_2$. On the other hand, the model suggests that the $RO_2$ + $RO_2$ pathway played a significant role in the chemistry of F26. This results, in part, from the higher initial NMOG loading for F26 ($\sim$ 280 ppb) compared to that of F38 ($\sim$ 180 ppb), which enhanced the rate of $RO_2$ production. The model suggests that the relative contribution of $RO_2$ cross-reactions was lower for higher carbon species; however, it is possible that these reactions produced accretion products unlikely to be found under ambient conditions. As discussed in Section 2.3.2 and shown in Figure S2, F26

was not representative of most fires studied here. For the majority of fires presented in Figures 2 and 3, the secondary NMOG were likely formed through the pathways consistent with F38.

In general, most of the higher carbon species in F38 followed atmospherically-relevant pathways. For some species, $RO_2$ + $RO_2$ reactions were also observed. While initial NMOG loadings may explain part of the enhanced $RO_2$ + $RO_2$ rate, some fraction may also be attributed to the limited degree of $NO_x$ cycling in the mini chamber. Under ambient conditions, $NO_2$

is photolyzed to NO, which then reacts with $RO_2$ (and $HO_2$) radicals. Consequently, the $RO_2$ + NO pathway may act to lower the fraction of radicals that follow the $RO_2$ + $RO_2$ pathway. This is evidenced by the biomass burning plume described





by Müller et al. (2016) which shows that the $RO_2$ + NO pathway is the dominant fate for most $RO_2$ species under ambient photolysis. In the mini chamber, $NO_2$ does not strongly absorb at 254 nm and is quickly lost to PAN or $HNO_3$; consequently, $NO_x$ does not play a significant role in the chemistry of higher carbon $RO_2$ radicals.

The primary focus of this study is to understand the formation of major secondary NMOG measured by I-CIMS; thus, it is instructive to identify the modeled radical pathways that contribute to these formation rates. Figure S14 shows the pathways for the radicals that lead to the formation of hydroxy furanone and methyl hydroxy furanone. For both experiments, the hydroxy furanone radicals predominantly react through $RO_2$ + $HO_2$ pathways. For F26, $\sim$ 15% of the $RO_2$ radicals react through the $RO_2$ + $RO_2$ pathways whereas $\sim$ 8% follow this pathway for F38. These results suggest that the $RO_2$ + $RO_2$ pathway played

some role in both experiments, but that the formation of hydroxy furanones in F38 predominantly followed atmospherically-relevant pathways. It is noted that as with other $RO_2$ radicals, these species are expected to mostly react with NO under ambient conditions (Fig. S14). Despite this difference, the $RO_2$ + $HO_2$ pathway and the $RO_2$ + NO pathway are both expected to lead to hydroxy furanone formation (Figures S4 and S5).

### 3.2.4    Observations and box modeling of secondary NMOG in real biomass burning plumes

Section 3.2.3 shows that the radical pathways in the mini chamber exhibited similarities, as well as differences, to those expected under atmospheric photolysis. In order to evaluate the impact of furan chemistry under ambient conditions, this work builds upon the model described by Müller et al. (2016) to evaluate the updated MCM mechanism with measurements from a real biomass burning plume.

    During the 2013 DISCOVER-AQ aircraft campaign, the NASA P-3B conducted several plume intercepts downwind of a
controlled burn conducted in a mixed-forested ecosystem. NMOG were monitored by a PTR-ToF-MS, meteorological parameters (temperature, pressure, and relative humidity) were monitored by a suite of aircraft instrumentation, and $NO_x$ and $O_3$ were monitored by chemiluminescence. Müller et al. (2016) modeled the chemical evolution NMOG using a semi-Lagrangian box model with a modified-MCM mechanism that included a basic oxidation scheme for furan and furfural. The authors successfully modeled the loss of primary NMOG, including furan and furfural, and captured trends in ozone, $NO_x$, and peroxyacetyl
nitrate (PAN). Downwind of the fire, the authors observed the formation of NMOG, such as maleic anhydride, that could not be explained by the model.

    The authors initialized the chemistry with measurements of NMOG, $NO_x$, and $O_3$ sampled in close proximity of the fire. Plume dilution was constrained based on the temporal evolution of CO. Background NMOG, $NO_x$, and $O_3$ concentrations were prescribed based on aircraft measurements conducted outside of the plume. The plume was simulated for 1 hr, and mete-
orological parameters were constrained based on measurements conducted at each plume crossing. Photolysis was prescribed based on observed $NO_2$ photolysis rates.

    The analysis described Müller et al. (2016) is recreated here, but with the full mechanisms of furan, furfural, 2,-methylfuran, 2,5-dimethylfuran, and 5-methylfurfural incorporated into the MCM. Other mechanisms that were not previously analyzed by Müller et al. (2016) are also considered, including phenol, cresol, and catechol. These species have important contributions to
the primary OH reactivity of biomass burning smoke and are explicitly represented in MCM v 3.3.1 (Fig. 4). The initial condi-





tions of furan and furfural are prescribed based on the observed concentrations reported by Müller et al. (2016). 2-methylfuran, phenol, and cresol are constrained based on the signals measured at $m/z$ 83.05 ($C_5H_6O-H^+$), $m/z$ 95.045 ($C_6H_6O-H^+$), and $m/z$ 109.066 ($C_7H_8O-H^+$), respectively. 2,5-dimethylfuran is isobaric with furfural and was not fully resolved by PTR-ToF-MS due to the overwhelming signal of furfural ($\sim$ 10 times greater than the signal of other furan species). 2,5-dimethylfuran is

expected to significantly contribute to total NMOG concentrations and OH reactivity (Fig. 4); therefore, the initial concentration of 2,5-dimethylfuran is constrained based on the dimethylfuran/methylfuran ratio reported elsewhere ($\sim$ 0.5, Koss et al., 2018). Finally, methylfurfural and catechol are included based on the signal at $m/z$ 111.049 ($C_6H_6O_2-H^+$), and it is assumed that 50% of the signal can be attributed to each compound as recommended by Koss et al. (2018). The initial concentrations of methylfurfural + catechol and cresol are adjusted to best match the decay of $C_6H_6O_2-H^+$ and $C_7H_8O-H^+$, respectively.

Figure 10 compares the model output to the dilution corrected concentrations of furans, oxygenated aromatics, maleic anhydride, and ozone. The model output of hydroxy furanone is also shown, but not compared to measurements since an I-CIMS was not onboard the P-3B. Red lines show the model output with furan chemistry included in the model. Blue dotted lines show model output with initial furan concentrations set to zero.

The model satisfactorily reproduces the temporal profiles of furans, oxygenated aromatic species, maleic anhydride, and

ozone. The model also predicts significant formation of hydroxy furanone. Similar to the mini chamber observations (Fig. 1), maleic anhydride exhibits a temporal profile that is consistent with a slow-forming product. The model output of hydroxy furanone exhibits a fast-forming temporal profile as expected from the mini chamber experiments (Fig. 1).

The production of maleic anhydride and hydroxy furanone is negligible when the initial concentrations of furan species are set to zero. For both species, furfural oxidation accounts for more than 50% of the total production. As discussed in

Section 3.2.1, the furfural mechanism is based on theoretical calculations and the exact branching ratios may differ from those estimated by Zhao and Wang (2017). The assumed branching ratios weakly impact the formation of maleic anhydride, whereas hydroxy furanone is most impacted by the assumed branching ratio of the ring-retaining pathway (channel B, Fig S16). I-CIMS measurements of hydroxy furanone may provide better constraints on the relative importance of each pathway.

Good agreement between ozone measurements and model output was also observed by Müller et al. (2016). Most of the

ozone production results from reactions involving $HO_2$ (formed primarily from OH + formaldehyde, CO, and furfural reactions), $CH_3O_2$ radicals (formed primarily from reactions involving acetaldehyde, 2,3-butanedione, and methylglyoxal), and NO. When furans are removed from the model, predicted ozone concentrations decrease by 12%. It is estimated that $\sim$ 5 ppb of ozone was produced from furan chemistry after 60 minutes of oxidation. Notably, ozone formation was not sensitive to the assumed furfural branching ratios (Fig. S16). It is noted that the contribution of ozone from furan chemistry will vary

depending on $NO_x$ conditions and that this estimate is not equivalent to a generalized ozone formation potential.

The products of furan chemistry are also reactive (e.g., hydroxy furanone, 1,4-butenedial, methyl hydroxy furanone, Figs S4-S9) and 1 hr of oxidation is too short to capture the total potential ozone produced from the oxidation of furan precursors. Figure 11 extrapolates the model forward to evaluate multi-generational oxidation processes on ozone formation. The model is extrapolated assuming that the dilution rate continues to follow an exponential decay (calculated based on the measured CO





loss). The solar zenith angle and $j_{NO2}$ are calculated based on time of day. Relative humidity, temperature, and pressure are assumed to remain constant following the last measured plume intercept.

    Figure 11 shows the extrapolated modeling results of ozone, hydroxy furanone, and maleic anhydride until 5:30 PM local time when the solar zenith angle approaches 70° ($\sim$ 1.5 hr before sunset). Hydroxy furanone production maximizes after 1 hr of oxidation, and subsequently decays due to OH oxidation. In contrast, maleic anhydride continues to increase. Figure 11A

shows the estimated ozone produced from furan chemistry (calculated as the difference between model runs initialized with and without furan species). Ozone production from furan oxidation continues to rise after 1 hr of aging, in part because of the oxidation of reactive secondary NMOG such as hydroxy furanone. After 4.5 hr of oxidation, the total ozone produced from furan chemistry is $\sim$ 8 ppb.

    Figure 11 demonstrates that furan chemistry contributed to the evolution of ozone within 4 hours of emission. After 2 hr

of aging, most furans have reacted ($<$ 20% remain), and their contribution to ozone production via reactions of secondary NMOG diminishes. It is important to note that ozone production will vary depending on $NO_x$ availability, $NO_x$/NMOG ratios, the chemical composition of the NMOG mixture, and meteorological conditions. Despite these factors, furan chemistry will likely play a role in ozone production for many biomass burning plumes due to the ubiquitous presence of furans in smoke (Stockwell et al., 2014, 2015; Gilman et al., 2015; Hatch et al., 2015; Bruns et al., 2017; Hartikainen et al., 2018; Koss et al.,

2018; Sekimoto et al., 2018).

    The plume described above is relatively young; however, observations from the mini chamber (Fig 1) and the continued formation of maleic anhydride in the extrapolated model suggests that this compound could be present in highly aged plumes. During the NOAA Shale Oil and Natural Gas Nexus (SONGNEX https://www.esrl.noaa.gov/csd/projects/songnex/) field campaign, the NOAA WP-3D aircraft intercepted a large biomass burning plume in the free troposphere above Montana, U.S. on

April 21, 2015 (Baylon et al., 2017). The plume had been transported at least 4 days from wildfires in Siberia and affected large portions of the western U.S. The PTR-ToF-MS described in this study was also deployed on the WP-3B and the resulting measurements are presented in Fig 12. The aged plume (indicted by elevated mixing ratios of acetonitrile and acetic acid) exhibited clear enhancements in maleic anhydride and $C_4H_4O_3$, which is attributed to succinic anhydride. In contrast, furan mixing ratios did not increase above background levels, indicating that these species completely reacted before sampling by

the P-3.

    The lifetimes of maleic and succinic anhydride are long ($>$ 5 d at OH concentrations of $1.5 \times 10^6$ molecules cm$^{-3}$); consequently, these species may have formed from the OH oxidation of furans and oxygenated aromatics shortly down wind of the fire and survived transport to the western U.S. These species may have also formed during transit from the oxidation of slow-reacting aromatics, such as benzene. The detection of anhydrides in highly-aged plumes is consistent with the behavior of

the mini chamber (Fig 1) and demonstrates the relevance of furans and aromatic oxidation on plume chemistry far downwind of fire sources.





## 4   Conclusions

This study evaluates the influence of understudied NMOG chemistry on the chemical evolution of laboratory and ambient biomass burning smoke. Smoke reacted with OH radicals exhibits fast decay of highly reactive organic species, such as furans and oxygenated aromatics, and significant formation of $C_4$ and $C_5$ oxygenates. A model incorporating furan chemistry into the MCM (v. 3.3.1) indicates that furan and oxygenated aromatic species are significant precursors of the secondary NMOG measured by PTR-ToF-MS and I-CIMS. Similar results are observed from modeling of a small understory fire (Müller et al., 2016), which demonstrates the importance of furan chemistry in real biomass burning plumes.

Maleic anhydride ($C_4H_2O_3$) and succinic anhydride ($C_4H_4O_3$) are identified as important secondary NMOG measured by PTR-ToF-MS. Those measured by I-CIMS are identified as methyl hydroxy furanone ($C_5H_6O_3$) and a mixture of hydroxy furanone and succinic anhydride ($C_4H_4O_3$). The link between these species and furan precursors may be useful in constraining OH exposures for biomass burning plumes measured in the ambient.

Previous studies have suggested that furan chemistry could play a significant role in ozone or SOA formation (Bruns et al., 2017; Hartikainen et al., 2018; Gilman et al., 2015; Hatch et al., 2015). For the biomass burning plume described by Müller et al. (2016), furan species likely contributed up to ~10% of total ozone production. The extent to which furans contribute to ozone formation will vary depending on burn conditions and models should include these species in order to improve ozone predictions, especially for young biomass burning plumes. Other major biomass burning oxidation products, such as formaldehyde and acetaldehyde, were not resolved by this work owing, in part, to the complexity of chemistry leading to small oxygenate formation. The unknown precursors or chemical processes leading to the formation of these species should be investigated in future work.

*Author contributions.*  Firelab data were measured and processed by the following people: PTR-ToF-MS (MMC, ARK, KS, JdG, and CW); ACES (KZ,SSB); I-CIMS (BY, JEK, and JLZ); OP-FTIR (VS, RY); GC/PTR-ToF-MS (ARK, JBG); NO (JMR). JK and CC designed the mini chamber experiments. CY and DHH operated the mini chamber. Ambient VOC measurements from the 2013 DISCOVER-AQ campaign were provided by MM and AW. MMC updated the MCM and conducted the model runs. MMC prepared the manuscript with contributions from all co-authors.

*Competing interests.*  The authors declare no competing interests

*Acknowledgements.*  This work was supported by the National Oceanic and Atmospheric Administration Atmospheric Chemistry, Carbon Cycle & Climate Program, awards NA16OAR4310111 and NA16OAR4310112. CL and ARK were additionally supported by the National Science Foundation Graduate Research Fellowship Program. The authors thank all those who helped organize and participated in the 2016 FIREX intensive, particularly Edward O'Donnell and Maegan Dills for lighting the fires, Ted Christian, Roger Ottmar, David Weise, Mark





Cochrane, Kevin Ryan, and Robert Keane for assistance with the fuels, and Shawn Urbanski and Thomas Dzomba for logistical support. The authors also thank Denise D. Montzka, David J. Knapp, and Andrew J. Weinheimer for permission to use $NO_x$ and $O_3$ measurements from the WP-3D, and John D. Barrick for permission to use $j_{NO2}$ measurements.

Data from mini chamber experiments and the SONGNEX flight from April 21, 2015 are available online (https://www.esrl.noaa.gov/csd/
datasets.html)



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

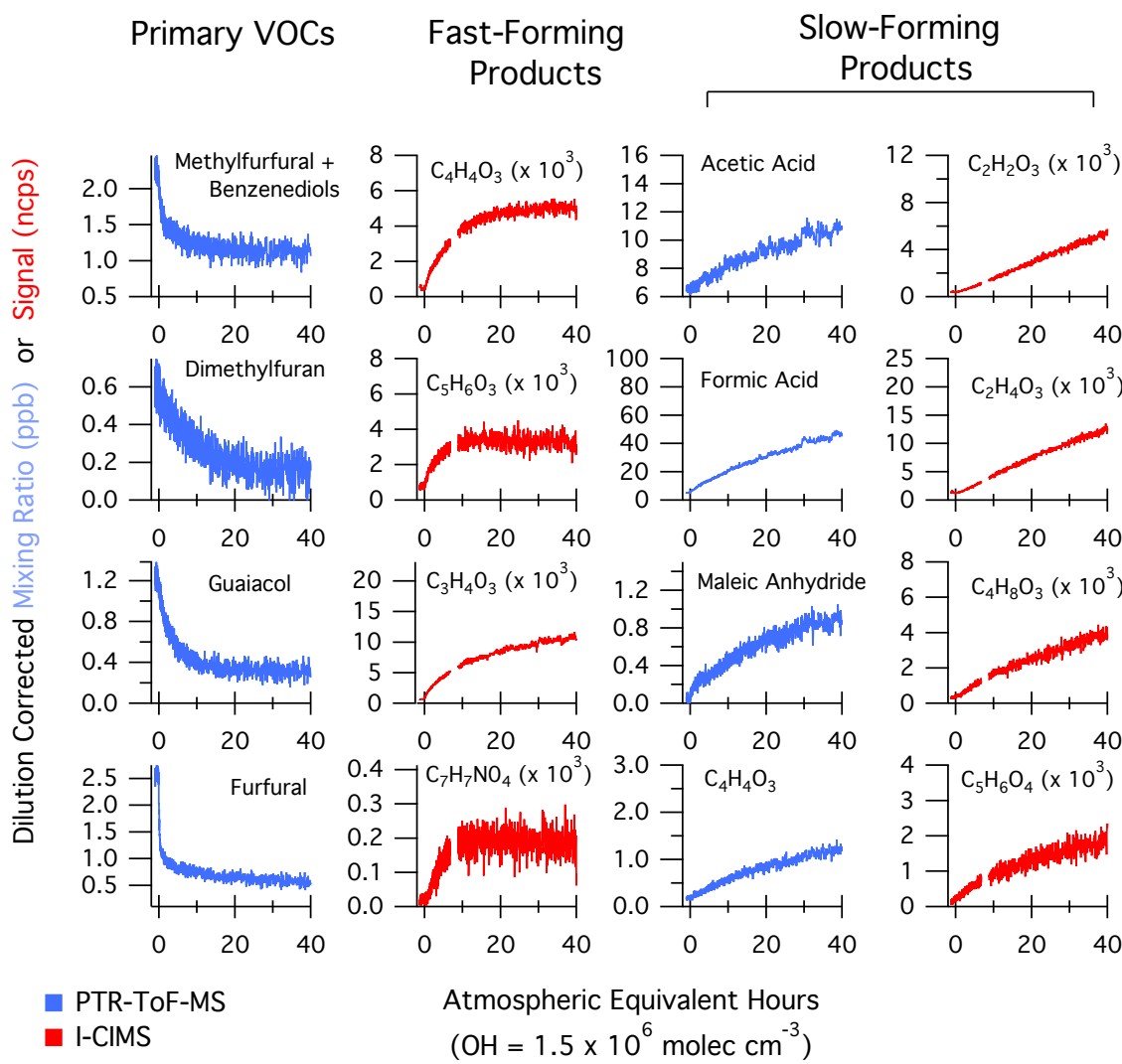

**Figure 1.** Temporal evolution of select NMOG from the OH oxidation of smoke resulting from Fire 38 (Ponderosa Pine). The data have been dilution-corrected based on the decay of acetonitrile. The time basis is calculated based on the decay of deuterated butanol. The first column shows primary NMOG, the second column illustrates species formed within 20 hrs of atmospheric-equivalent oxidation ("fast-forming" products), while other columns show species formed over longer time scales ("slow-forming" products).

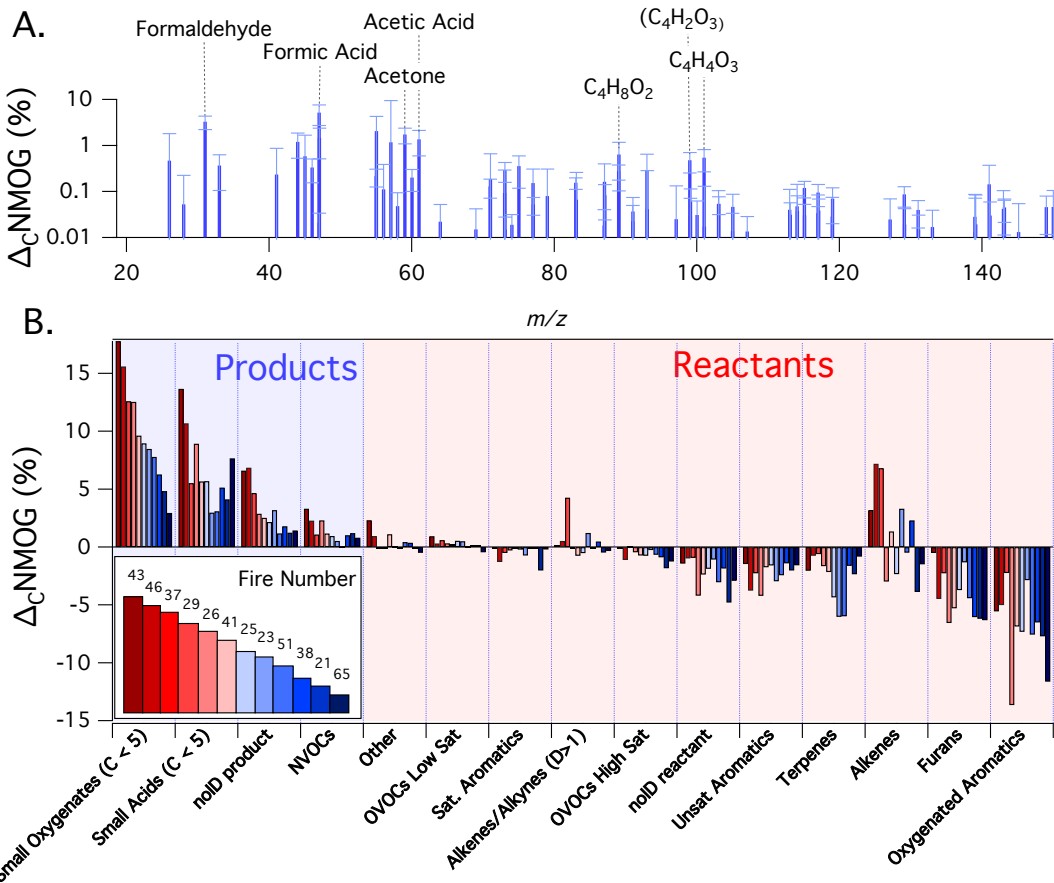

**Figure 2.** Summary of PTR-ToF-MS measurements during 12 mini chamber experiments. Shown is the amount of initial NMOG carbon ($\Delta_C$NMOG, %) that was consumed (reactant) or formed (product) for a given species after 12 hours of atmospheric-equivalent oxidation (OH concentration of $1.5\times10^6$ molecules cm$^{-3}$). Panel (A) shows the average speciated distribution of products (error bars are standard deviations). Panel (B) shows $\Delta_C$NMOG for reactants and products as lumped categories. A species is classified as a product if the dilution-corrected signal measured by the PTR-ToF-MS increased after 12 hours of simulated oxidation, and a reactant if the signal decreased. Categories labeled as "no ID" refer to species for which an assignment could not be confidently prescribed. The numbers in the legend refer to the fire number (composition and initial conditions can be found in Fig. S2). Experiments are ordered according to the fraction of primary carbon transformed to measurable secondary NMOG (greatest to least).

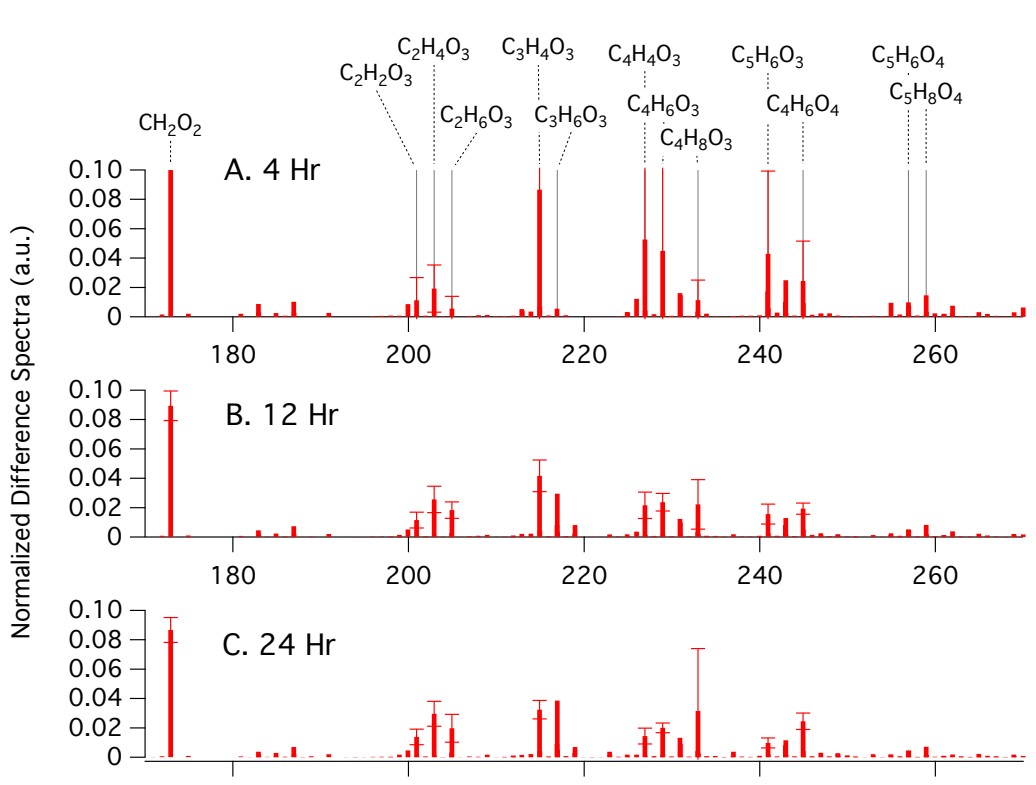

**Figure 3.** Summary of secondary NMOG measured by I-CIMS during 12 mini chamber experiments. Shown are ion signals that increased after initiating photochemistry relative to the total integrated signal measured prior to photochemistry. Panels (A)-(C) show changes in signal after 4, 12, and 24 hrs of atmospheric-equivalent oxidation, respectively. Error bars represent standard deviations. Note that the masses are presented as $I^-$ adducts.

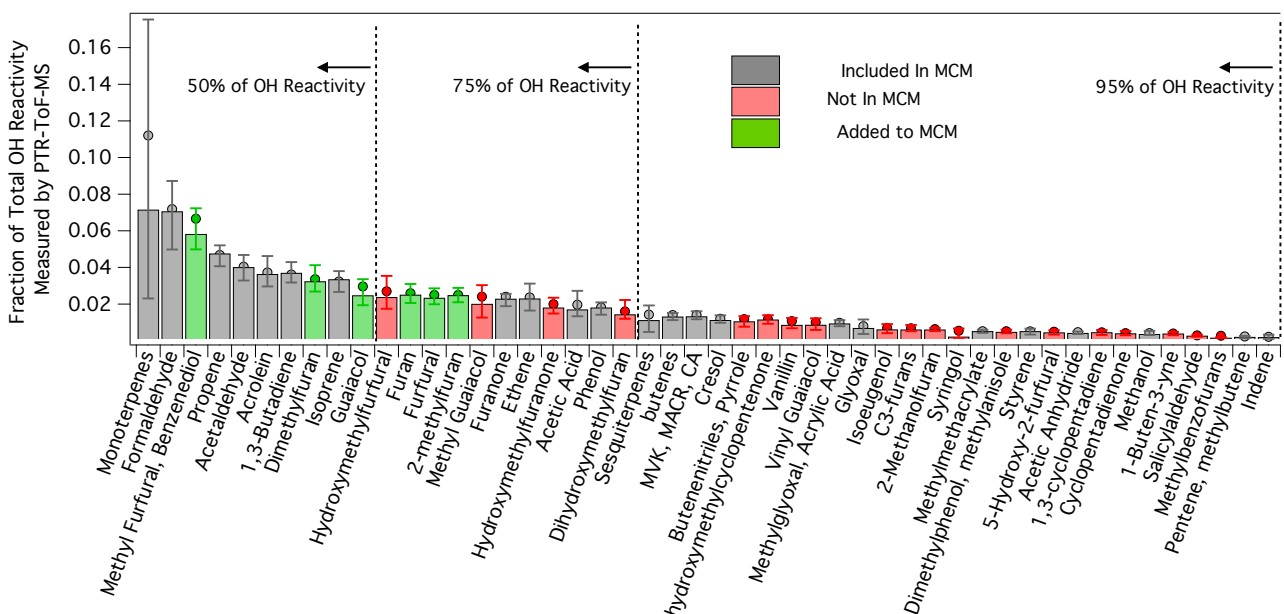

**Figure 4.** Contribution of individual species to the total primary OH reactivity estimated from PTR-ToF-MS measurements following Equation 2. Bar intensity is median for all burns measured by PTR-ToF-MS, whereas circle markers are averages. Error bars indicate $25^{th}$ and $75^{th}$ percentiles. Bar colors indicate if a species is included (grey) or not included (red) in the MCM v. 3.3.1. Green bars indicate OH oxidation mechanisms that were added to the MCM (sub-mechanisms for each species can be found in Figures S4-S9). The concentrations of each species were calculated following the methods described by Koss et al. (2018). Only species that were positively identified are included in these calculations. The assumed OH rate constant for each species is given by Koss et al. (2018).



**Figure 5.** Summary of OH oxidation pathways and final products for furan, 2-methylfuran, and 2,5-dimethylfuran where branch A is the substituent-retention pathway, and branches B1 and B2 are substituent-loss pathways (adapted from Aschmann et al., 2014). In this study, the branching ratios are assumed to be [0.7 (A), 0.3 (B1)] for furan, [0.31 (A), 0.39 (B1), 0.31 (B2)] for 2-methylfuran, and [0.27 (A), 0.73 (B1)] for 2,5-dimethylfuran. Details of the assumed $RO_2$ reaction schemes are provided in Figures S4 - S6.





**Figure 6.** Summary of OH oxidation pathways and final products for furfural and 5-methylfurfural, where branch A is the substituent-retention pathway, and branches B and C are substituent-loss pathways (adapted from Zhao and Wang, 2017). The branching ratios estimated by Zhao and Wang (2017) are 0.37 for channel A, 0.6 for channel B, and 0.03 for channel C. Details of the assumed $RO_2$ reaction schemes are provided in Figures S7 - S9.



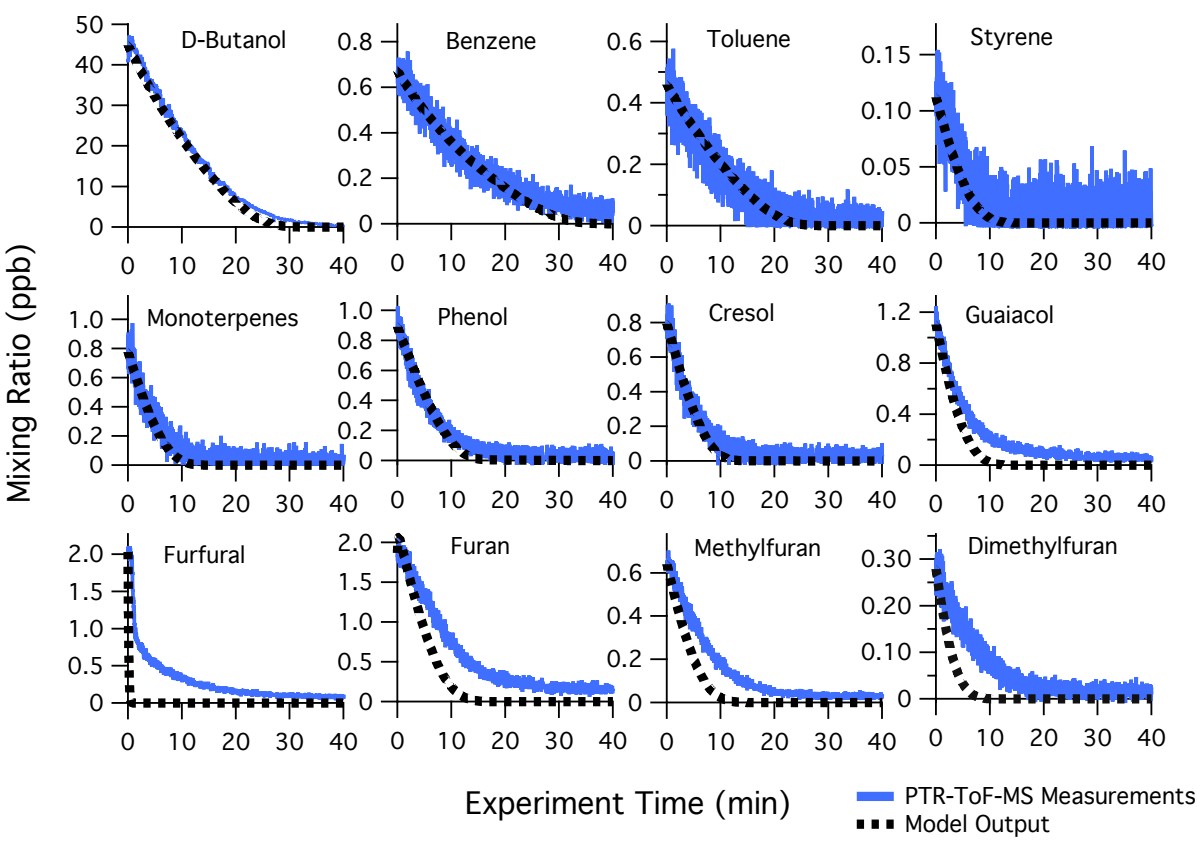

**Figure 7.** Primary NMOG measurements (blue lines) compared to modeled output (black dotted lines) for Fire 38. The decay of d-butanol is shown to demonstrate model performance in reproducing OH exposures, which was achieved by adjusted the measured photon flux by a factor of 1.5. Fuel = Ponderosa Pine Litter, $NO_x$/NMOG = 0.3, mixture of high and low-temperature pyrolysis products.



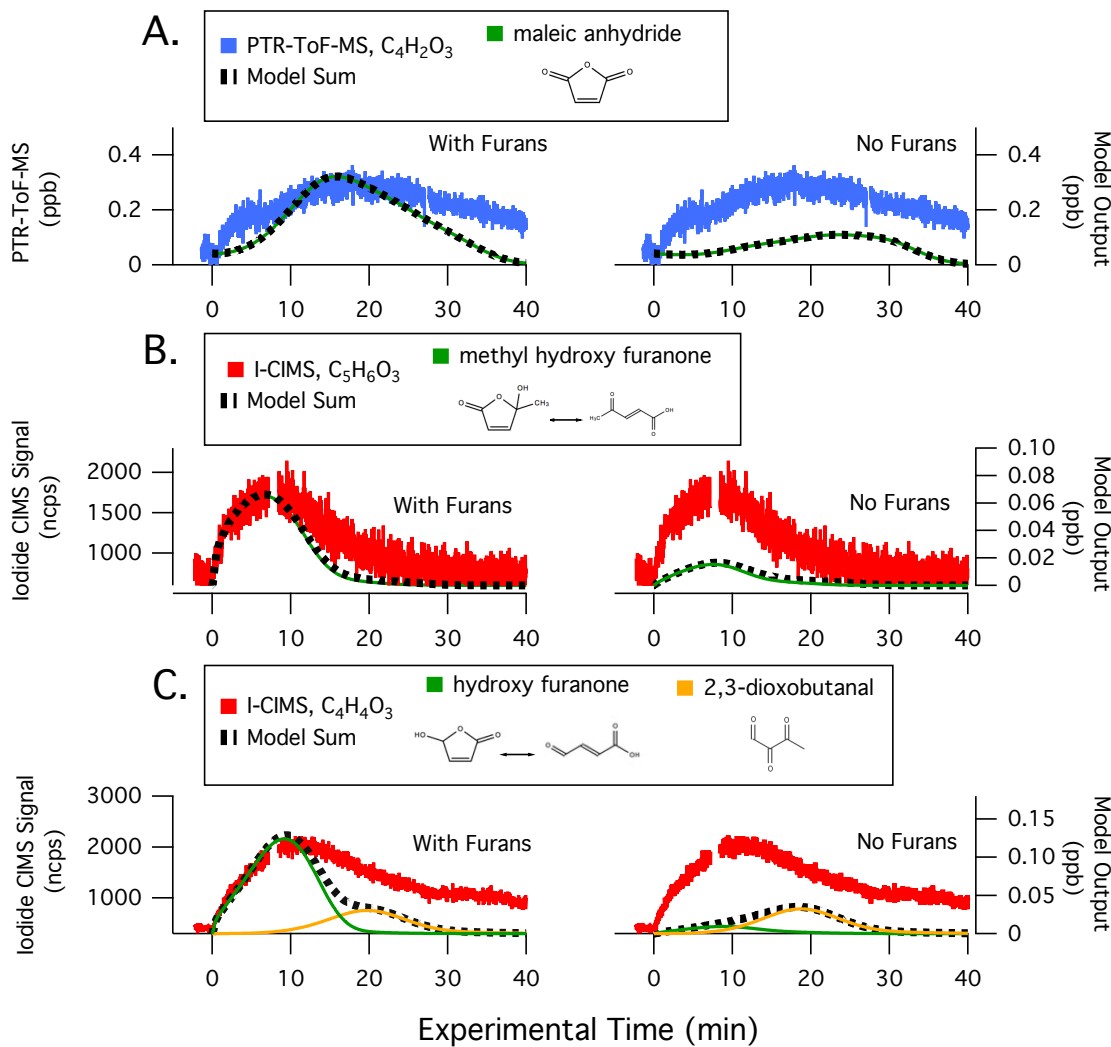

**Figure 8.** Secondary NMOG measurements compared to modeled output for Fire 38. Row (A) shows PTR-ToF-MS measurements of $C_4H_2O_3$ compared to model output of maleic anhydride. Row (B) shows $I^-$-ToF-CIMS measurements of $C_5H_6O_3$ compared to model output of methyl hydroxy furanone and its tautomer, $\beta$-acetylacrylic acid. Row (C) shows $I^-$-ToF-CIMS measurements of $C_4H_4O_3$ compared to model output of hydroxy furanone, its tautomer malealdehydic acid, and 2,3-dioxobutanal. All graphs to the left show full model runs, while graphs to the right show model runs when the initial conditions of furan, 2-methylfuran, 2,5-dimethylfuran, furfural, 5-methylfurfural, and furanone are set to zero.



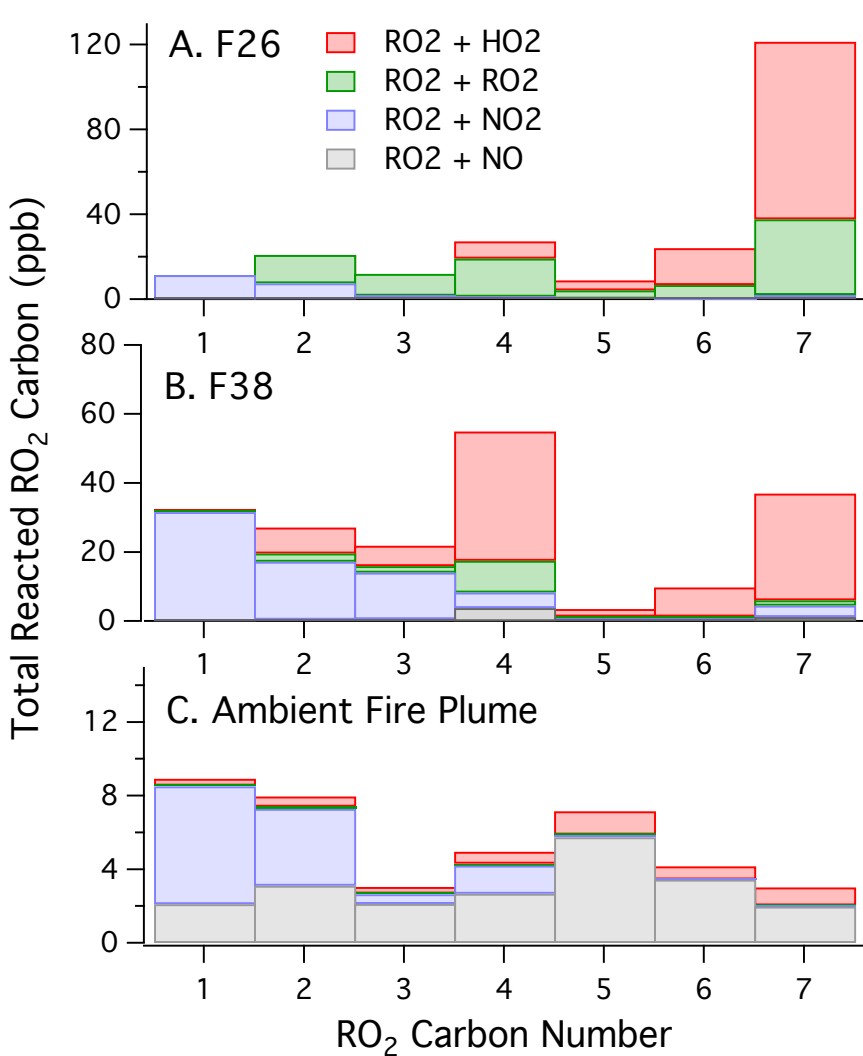

**Figure 9.** The fate of $RO_2$ species for (A) F26, (B) F38, and (C) the ambient biomass burning plume described by Müller et al. (2016). $RO_2$ species are grouped by carbon number, and the color of each bar shows the fraction of $RO_2$ species reacted by $RO_2 + HO_2$, $RO_2 + RO_2$, $RO_2 + NO_2$, and $RO_2 + NO$ pathways.

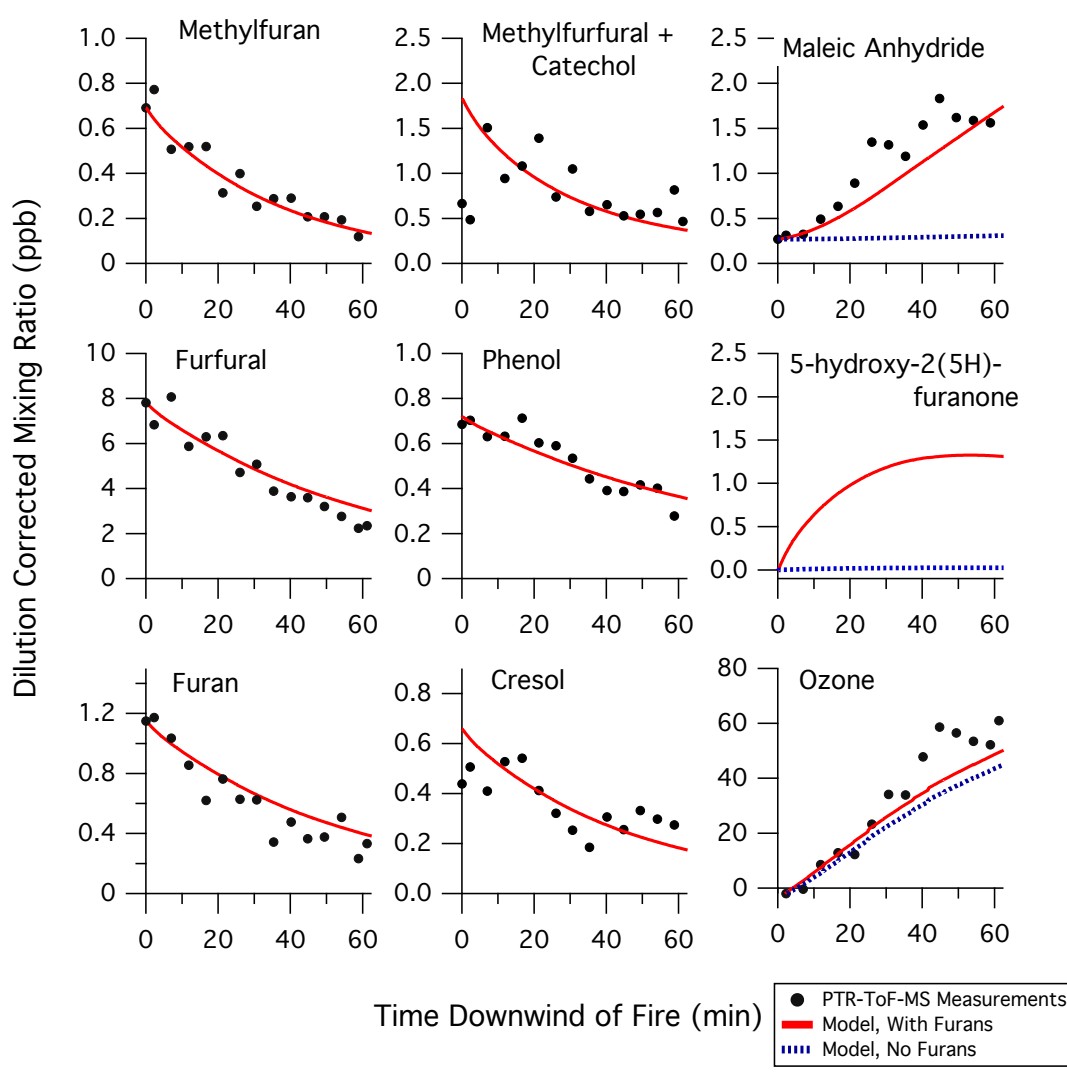

**Figure 10.** Summary of model results for the understory fire described by Müller et al. (2016). Data are presented as mixing ratios corrected for dilution (calculated based on the decay of CO). The initial conditions for methylfurfural + catechol and cresol were adjusted to best match the observed decay. Results with furans excluded from the model are shown as dotted lines. Maleic anhydride and ozone are shown to illustrate the impact of furans on secondary product formation.

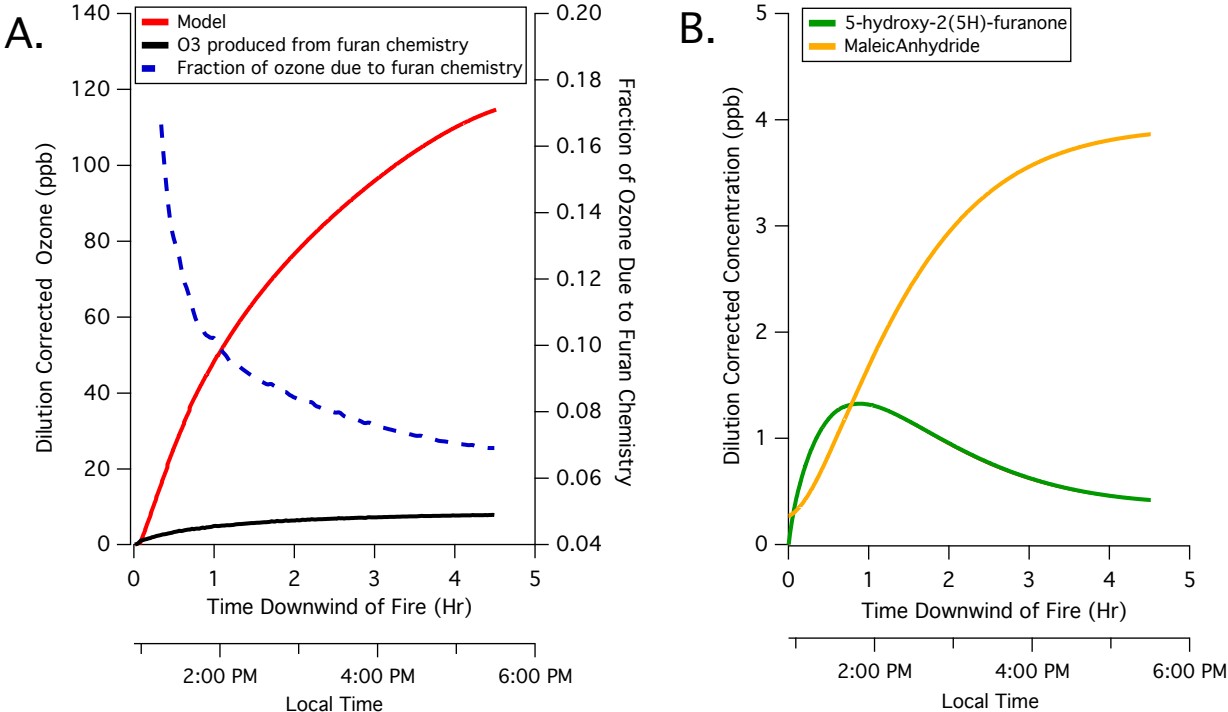

**Figure 11.** A. Total ozone, estimated ozone produced by furan chemistry, and fraction of total ozone associated with furan chemistry for the modeled biomass burning plume measured by Müller et al. (2016). B. Model predictions of 5-hydroxy-2(5H)-furanone and maleic anhydride production. Shown is an extrapolation of the 1 hr model described in Section 3.2.4. After 10 minutes of oxidation, furan chemistry is responsible for > 15% of total ozone production in the model. The predicted contribution from furan chemistry decreases as furans are consumed and other, slower-reacting primary NMOG begin to oxidize. Furan chemistry contributes ∼ 8 ppb of ozone after 4.5 hr of oxidation.

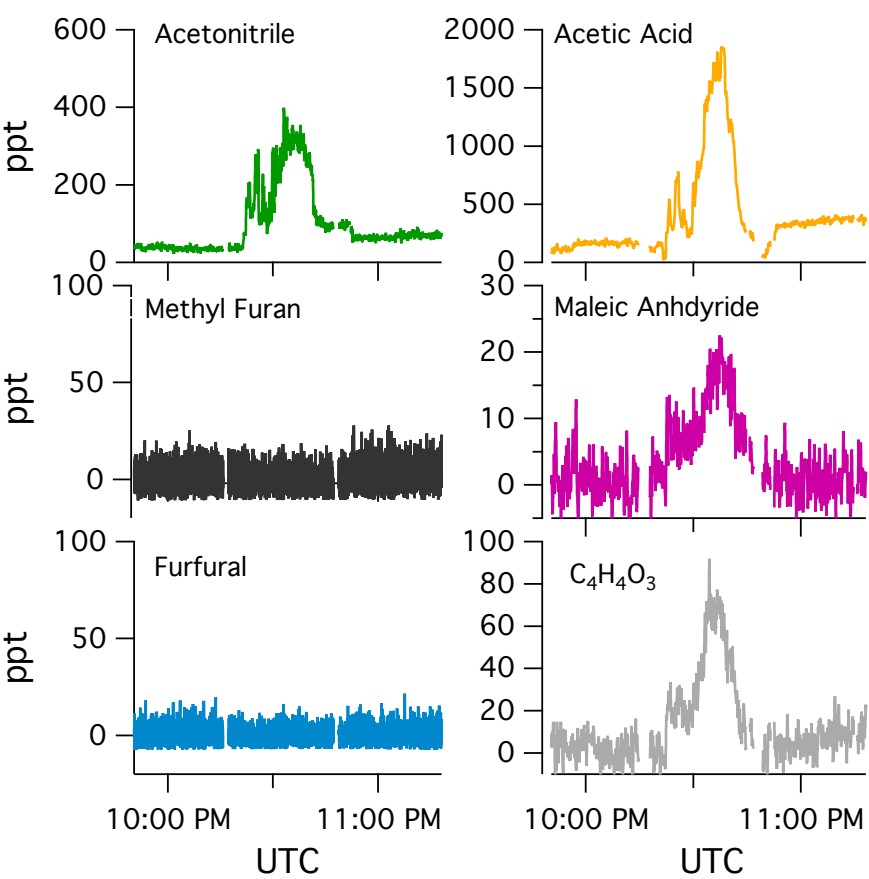

**Figure 12.** PTR-ToF-MS measurements of a wild fire plume transported to the U.S. from southern Siberia. The plume was intercepted April 21, 2015 during the SONGNEX field campaign and is described in detail by Baylon et al. (2017).