# Peer review of "OH-chemistry of non-methane organic gases (NMOG) emitted from laboratory and ambient biomass burning smoke: evaluating the influence of furans and oxygenated aromatics on ozone and secondary NMOG formation."

_Atmospheric Chemistry and Physics, 2019_

## Referee Comment (RC1) · Anonymous Referee #1 · 17 Jul 2019

General comments:

This paper investigates the role of a number of identified furans and other oxygenated aromatic compounds in the chemistry of ageing biomass burning emissions. Chemical schemes are developed for these species, and are used to extend the chemistry in the Master Chemical Mechanism (MCM). The extended mechanism is used to interpret the results of simulation experiments in which biomass burning smoke is oxidized in a

chamber under illumination from UVC lamps; and the description of secondary organic compound formation (measured by PTR-ToF-MS) is reported to be improved. The mechanism is also tested in a Lagrangian box model used previously to model a real biomass burning plume. This is reported to provide an improved representation of the chemistry in biomass plumes, in particular the formation of maleic anhydride which may be used as a tracer for this source.

This is an interesting piece of work, providing results and interpretation that will potentially help to improve the understanding and representation of the chemistry of ageing biomass burning emissions. The work is suitable for publication in ACP, but there are a number of points where additional clarification and information would seem to be required. The authors should address the following comments in producing a revised version of the manuscript.

Specific comments:

Lines 107-115: Some information is given here on primary OH radical sources in the "mini-chamber", and more generally in Sect. 3.2.3 and the SI about photolysis rates using the UVC lamps. MCM v3.3.1 recommends sources of cross-section and quantum yield data in relation to the specific and generic photolysis processes it represents (http://mcm.leeds.ac.uk/MCMv3.3.1/parameters/photolysis.htt). Can the authors confirm that these were used, or provide more details on what information was used (e.g. in Table S1 for emitted compounds and elsewhere for product species, e.g. carbonyls, hydroperoxides and nitrates)?

Line 122. It is stated that "Gas-phase species have a high affinity to metal surfaces...." Acetonitrile is a gas phase species, but is assumed not to be lost to surfaces (stated in the SI). Perhaps a little more information on this is required in the manuscript, including a reference to support the acetonitrile assumption.

Line 129: "Deuterated butanol" is abbreviated as "d-butanol". Neither of these terms tells the reader what the molecule actually is, and more clarification is required. Indeed, the prefix "d-" (as opposed to "l-") usually specifies the rotation direction of optical activity... and this is not relevant in this case. Assuming "deuterated butanol" is the same species for which Barmet et al. (2012) measured the OH rate coefficient (line 139), it is CD3CD2CD2CD2OH. This should be identified as "1-butan-d9-ol" or "butanol-d9".

Line 265: A little more explanation could be provided on the deviations between chamber and ambient chemistry that occur at high OHR, and why values of 70 s-1 and below are acceptable.

Line 390: The use of the MCM a-pinene mechanism alone to represent all monoterpenes seems unnecessarily simplistic. As stated, the MCM includes a-pinene, b-pinene and limonene as examples of monoterpenes respectively containing endocylic-, exocyclic- and endocyclic- + exocyclic- double bonds, and therefore has significant coverage of the list given previously on line 390. An improved approach would surely have been to represent those three monoterpenes explicitly, and use them as mechanistic surrogates for the remaining ones. For example, the OH- and O3- initiation reactions for camphene (containing an exocyclic- double bond) could be included explicitly in the mechanism, but represented to form the corresponding products in the b-pinene scheme (this being the closest structural representative).

It should also be noted that the MCM includes the chemistry of the sesquiterpene, b-caryophyllene. Presumably, sesquiterpenes are also produced from biomass burning?

Line 423: Hydroperoxide photolysis is a slow process in the atmosphere, even if much more rapid with 254 nm illumination in the chamber. What photolysis rates are used? Reaction with OH will almost certainly dominate in the atmosphere. Have you investigated the effect of assuming a radical forming process (photolysis) in place of a radical propagating process (OH reaction) in the ambient plume simulations?

It would be a very helpful service to the community if the added chemistry was also provided in the form of a reaction listing that can be used in conjunction with MCM

v3.3.1. It should also be made clear that expressions like "the updated MCM" used at various points in the manuscript do not imply that the MCM itself has been updated. Perhaps "the extended MCM" or "the customized MCM" would be more appropriate.

Line 425: kRO2HO2 actually has a size-dependence term in the MCM, based on carbon number (see Saunders et al., 2003). The quoted rate constant is the limiting value at high carbon number. Was this taken into account?

Line 427: Although MCM v3.3.1 does not generally contain RO2 H-shift isomerization reactions, it does contain those specifically identified to occur during the degradation of isoprene (see Jenkin et al., 2015), which is included in the speciation applied in this study. Perhaps this statement should be qualified accordingly.

Line 457: If butanol-d9 is a significant contributor to OH reactivity, this approach would seem over simplified. Fig. 7 suggests use of 45 ppb, equating to and OH reactivity of about 3.8 s-1 - compared with about 2.3 s-1 for 5-methylfurfural and 2-methylfuran collectively. If this latter pair is significant (as stated on line 399) then butanol-d9 is even more significant and it would seem prudent to represent its chemistry more explicitly.

There is also the possibility that its degradation might form products of masses that interfere with those from the studied oxygenated compounds, and this should be checked. Based on a quick appraisal, I think one product type (C4H2D8O3 - i.e. CD3CD2CD(OOH)CD2OH and CD3CD(OOH)CD2CD2OH) is formed in the first generation of oxidation of butanol-d9, from RO2 + HO2 reactions. This has an interfering mass of interest (114), which I believe is isobaric with methyl hydroxy furanone and beta-acetylacrylic acid.

Line 498: Presumably, the slower decay might also result from formation of isobaric products in the system?

Line 528: Statements like "....chemical pathways that are unaccounted for within MCM v 3.3.1" are very easy to include as explanations, but are often not backed up by evi-

dence or specific examples. The degradation of almost all VOCs produces formalde-hyde, and this is actually generally found to be substantial and quite well represented in MCM. For example, looking at your list of species, the 3 most abundant (after formalde-hyde itself) are 1-butene, methanol and ethene, with these making a collectively signif-icant contribution to the OH reactivity. 1-butene and methanol generate formaldehyde essentially quantitatively (on a molar basis) from radical propagation pathways in the first oxidation step, with ethene producing about 1.7 molecules of formaldehyde. MCM represents this very well, and continues to represent formaldehyde formation from the further oxidation of C2 and higher products (e.g. C2H5CHO from 1-butene). Going on down the list, most of the species make formaldehyde from first- and/or higher-generation chemistry in MCM v3.3.1 - but with the obvious exceptions being species that do not contain a -CH2- substructure (e.g. ethyne, formic acid, glyoxal, phenol and benzene) which cannot make formaldehyde. The much more likely explanation for the under-representation of formaldehyde formation is therefore missing species in the starting speciation. It might also be very sensitive to uncertainties in the [NO] initializa-tion in the model, as this would influence the relative importance of radical propagation and radical termination processes.

Line 603: MCM only includes RO2 + NO2 for acyl peroxy radicals and CH3O2. The reaction is not represented in most cases because the RO2NO2 products are ther-mally unstable and rapidly regenerate RO2 and NO2 (as indeed the CH3O2 + NO2 reaction does at 298 K). The RO2 + NO2 contribution is therefore probably dominated by a very limited set of reactions. The C1 reaction is only CH3O2+NO2, C2 is proba-bly dominated by CH3C(O)O2 + NO2, C3 by C2H5C(O)O2 + NO2: and the small C7 contribution is probably mainly C6H5C(O)O2 + NO2. Figure 9 therefore suggests that this limited set of reactions has a big effect. It might also be more correct to check the back decomposition rates for RO2NO2 = RO2 + NO2 and to present the net (forward minus reverse) effect of the RO2 + NO2 reaction - particularly for CH3O2 + NO2 = CH3O2NO2. Note also that the decomposition rates are very sensitive to tempera-ture. Was temperature measured in the mini-chamber and taken into account in the

simulations?

Typos:

Line 399: "2-dimethylfuran" should be either "2-methylfuran" or "2,5-dimethylfuran".

Line 586: "undesireable" should be "undesirable"

---

## Referee Comment (RC2) · Anonymous Referee #2 · 23 Jul 2019

Coggon et al., have used an extensive data set of PTR-ToF-MS and I-CIMS measurements, carried out during a wide range of chamber biomass burning /OH oxidation experiments in order to investigate the impact of a range of observed emitted furans and oxygenated aromatics on the formation of secondary reactive VOC and ozone in aged smoke. This was inferred from model/measurement comparisons of detailed chemical box modelling, where schemes from the Master Chemical Mechanism (MCMv3.3.1) were updated and extended. The results highlight significant formation of a range

of C4 and C5 anhydrides and furanone species in aged biomass burning plumes. The derived mechanisms were put into a Lagrangian box model in order to model the chemical evolution of a real life biomass burning again highlighted the importance of furan chemistry and its impact on ozone formation.

The work presented is scientifically and mechanistically interesting and the comprehensive compositional measurement dataset, mechanism development and model interpretation work significantly adds to our further understanding of the atmospheric impact of biomass burning on fundamental atmospheric chemistry and air quality. The work is very much suitable for publication in ACP, after the following comments and clarifications have been addressed.

Specific points

"mini-chamber" experiments and the use of UVC lamps as a large, "clean" source of OH

The burn smoke is diluted with humidified air (why 30%?)  and introduced into the mini-chamber along with ozone (why 70 ppb?), where the chamber reactive mixture is exposed to (predominantly?) 254 nm UVC radiation in order to produce OH to initiate the chemical degradation of the chamber mix. Therefore, the chemistry in the system is not carried out under atmospherically relevant conditions, these studies are aimed mainly at looking at the mechanisms of the OH chemistry. The chemistry is dominated by OH chemistry, with little information given on the NOx and NOy composition in the mini-chamber (only estimated from stack measurements) and any photolysis chemistry of photo-labile species is occurring at non-atmospherically relevant wavelengths.

There is little information given on the chamber auxiliary chemistry mechanism needed in order to the background reactivity of the chamber in the chamber specific box modelling. For example, processes that need to be taken into account include: introduction of free radicals from heterogeneous chamber wall reactions; adsorption/desorption of NOy species (including HONO) to/from the chamber walls; off-gassing of various reactive species from the chamber walls, which can contribute significantly to the radical budget of the system. Also, initial HONO concentration is important to know. Is a detailed chamber auxiliary mechanism available and used here? Formic and acetic acid are shown to be significant secondary product VOCs in the experiments (Line 288). These small oxygenates, as well as HCHO and HONO have significant wall sources in Teflon chambers. Have these sources been taken into account? (in the chamber auxiliary mechanism).

It is stated at line 238/239 that "radical reactions and NOx loss process are likely sensitive to initial NO/NO2 ratios", have any model sensitivity analyses been done to look at this sensitivity?

More chamber details are needed – mixing time, spectrum of the lamp used and how was the photon flux derived; how uniform throughout the chamber is the photon flux exposure?

As the other reviewer points out, information on the photolysis rates of some of the primary photo-labile VOCs, along with rate constants for other atmospherically important oxidants are given in the SI. However, little to no information has been given as to the sources of these important data are given. Also, how were the photolysis rates calculated? It is important to note that the MCM chemical mechanisms are designed for use in tropospheric chemistry models, where the cross-sections and quantum yield data used to derive the photolysis rates are mainly used at > 290 nm. Therefore, any primary and secondary MCM chemistry used here in the chamber modelling (but not in the BB plume models) needs to be adjusted to photolysis < 290 nm (mainly around 254 nm). For example, the 1,4-dicarbonyl chemistry (mainly 2-butenedial and 4-oxo-2-pentenal, which are significant products from the atmospheric chemistry of aromatics and furans) in the MCM is mainly driven by photolysis (see Newland et al., 2019), where the main photolysis products are furanones and maleic anhydride (a main focus of this study). However, at 254 nm, the photolysis product distribution (quantum yields and photo-tautomer distributions) are different (Tang and Zhu 2005).
[Figure]

I agree with the other referee that given that a) monoterpenes contribute significantly to the calculated OH reactivity on the experiments and b) the MCM contains extensive chemical mechanism for the atmospheric degradation of both alpha and beta pinene, as well as limonene (and beta caryophellene), using a-pinene as a proxy is an over-simplification here (Line 394).

Line 80 – define "0-D" Line 105 – define "semi-batch" Line 275 – why is an OH atmospheric concentration of 1.5E+6 cm-3 assumed? Line 353 – "I-ToF-MS" should be I-CIMS? Line 364 – "anthropogenic and biogenic aliphatic and aromatic species" Line 484 – reference needed for the o-semiquinone chemistry Line 501 – is this exhibiting a "bi-exponential" decay profile? Line 515 – could the model be underpredicting the MA yield owing to increased MA and furanone yields from 1,4-dicarbonyl photolysis at 254 nm? Line 537 and Figure 2 – under-prediction of small oxygenates – some of this could be coming from wall sources? (e.g. Zador et al., 2005) Line 560 – ozonolysis of other monoterpenes than a-pinene could be more important? Line 718 – "understudied NMOG OH chemistry" – this study really focuses on the OH chemistry Line 721 – what does "understory" mean?

Figure S6, S7 and S9 – J<41> (CH3OOH photolysis rate) – has this been adapted for 254 nm photolysis? Figure S10 – what is GUAIACOLPROD? Figure S14 – give structure of HYDMEFURANO2 on plot

M. Newland et al., Phys.Chem.Chem.Phys., 2019, 21, 1160. Y. Tang and L. Zhu, Chem. Phys. Lett., 2005, 409, 151–156. Zador, J., Turanyi, T., Wirtz, K., and Pilling M. J, J. Atmos. Chem. 55, 147–166, 2006.

---

## Referee Comment (RC3) · William Stockwell (Referee) · 25 Jul 2019

This paper provides some important insights into emissions from biomass burning.

Biomaterial was burned and chemical measurements of the resulting emissions. The MCM chemical mechanism was used in simulations of the experiments. The authors added more chemical reactions to the MCM to improve its performance in simulating the experiments.

[Figure]

Much more data like this is needed for the continuing development chemical mechanisms for atmospheric chemistry modeling. Also the authors have used some quantum chemistry in the further development of the test reactions that were added to the MCM. Finally, they considered real biomass burning plumes.

All in all, this paper represents a good mix of laboratory measurements, mechanism development with evaluation and applications to the real world. It is an excellent template for future studies.

Minor: Lines 210 - 215 I commend the authors for using the correct term, photolysis frequencies in this section. However, they need to be consistent and state at the beginning: "Photolysis frequencies are calculated using literature cross-sections and quantum yields of relevant organic and inorganic species..."
* * *

---

## Author Comment (AC1) · 16 Oct 2019

**Summary of Revisions**

We thank all the reviewers for these very helpful reviews. We appreciate the careful scrutiny of our mechanistic work, especially as it pertains to organic photolysis.

Reviewers 1 and 2 provided a number of important, overlapping comments. Before addressing each point, we would like to summarize the main changes to the manuscript. All changes made to the manuscript are highlighted in red text.

(1) As the reviewers point out, we did not clarify the sources that were used to evaluate photolysis frequencies. We apologize for this oversight. All of the photolysis parameters used in modeling the mini chamber were calculated using published cross-sections, known or estimated quantum yields, and the reported photon flux at 254 nm. We have added 2 tables in the supplemental information that summarize the sources used to calculate photolysis frequencies employed in the MCM (Table S1) as well as other reactions that were added to the mini chamber model to account for photolysis of primary species and NOx reservoirs (Table S2). In general, we use sources recommended by the MCM, unless these sources do not report cross-sections at 254 nm. The ambient model utilizes the MCM parameterizations relevant to ambient photolysis. We have also added text to the manuscript on pages 8 and 9 to describe these calculations.

(2) In light of revision (1), we have also clarified the purpose of the mini chamber model to acknowledge the uncertainties associated with photolysis. The primary reason for incorporating furan mechanisms into the MCM is for use in understanding VOC oxidation in ambient plumes. Our intention for modeling the mini chamber was to evaluate the likely identities of secondary NMOG detected by PTR-ToF-MS (e.g. maleic anhydride) and I-CIMS (e.g. hydroxyfuranones), and to show that furan chemistry plays an important role in this formation. We have added text on page 8 to explain the scope of the mini chamber analysis.

(3) We have included a FACSIMILE + readme file that includes the full mechanism used in this modeling - i.e., MCM 3.3.1 + furans.

(4) We have speciated the monoterpenes to account for differences in reactivity between exocyclic and endocyclic double bonds. We have also updated the RO2 + HO2 rate constants to account for carbon number. All of the graphs and tables have been updated to reflect these changes.

**Reviewer 1 Comments**

This paper investigates the role of a number of identified furans and other oxygenated aromatic compounds in the chemistry of ageing biomass burning emissions. Chemical schemes are developed for these species, and are used to extend the chemistry in the Master Chemical Mechanism (MCM). The extended mechanism is used to interpret the results of simulation experiments in which biomass burning smoke is oxidized in a chamber under illumination from UVC lamps; and the description of secondary organic compound formation (measured by PTR-ToF-MS) is reported to be improved. The

mechanism is also tested in a Lagrangian box model used previously to model a real biomass burning plume. This is reported to provide an improved representation of the chemistry in biomass plumes, in particular the formation of maleic anhydride which may be used as a tracer for this source. This is an interesting piece of work, providing results and interpretation that will potentially help to improve the understanding and representation of the chemistry of ageing biomass burning emissions. The work is suitable for publication in ACP, but there are a number of points where additional clarification and information would seem to be required. The authors should address the following comments in producing a revised version of the manuscript.

*We thank the reviewer for their careful evaluation of our mechanistic work. Please find our responses to these comments below.*

1) Lines 107-115: Some information is given here on primary OH radical sources in the"mini-chamber", and more generally in Sect. 3.2.3 and the SI about photolysis rates using the UVC lamps. MCM v3.3.1 recommends sources of cross-section and quantum yield data in relation to the specific and generic photolysis processes it represents. Can the authors confirm that these were used, or provide more details on what information was used (e.g.in Table S1 for emitted compounds and elsewhere for product species, e.g. carbonyls, hydroperoxides and nitrates)?

   *Thank you for this helpful comment. In general, we used the recommended sources from the MCM, unless these sources did not report cross sections at 254 nm, or higher resolution data were available. These sources and assumptions are provided in Tables S1 and S2, as described in the Summary of Revisions.*

2) Line 122. It is stated that "Gas-phase species have a high affinity to metal surfaces...."Acetonitrile is a gas phase species, but is assumed not to be lost to surfaces (stated in the SI). Perhaps a little more information on this is required in the manuscript, including a reference to support the acetonitrile assumption

   *Thank you for pointing this out - we could have been more clear about what we meant by this analysis. VOC losses to metal surfaces were studied by Deming et al. (2019), who showed that metal tubing effects are largely driving by volatility, functionality, and displacement processes at adsorption sites. However, all VOCs that were studied by Deming et al. (2019) had some delay through metal tubing. Our assumption was that acetonitrile is not lost to a greater extent than other species, given its higher volatility. In truth, we are not concerned by absolute losses (we acknowledge that assessing absolute losses is difficult), but rather relative differences in the NMOG distribution between the stack and chamber. The analysis in the SI shows that NMOG/acetonitrile ratios between stack and mini chamber were different (on average 15% lower in the mini chamber), but were not orders of magnitude different, which implies that the VOC distribution in*

*the mini chamber is reasonably representative of the mixture sampled from the stack.*

*We have clarified this in the main text, as well as in the SI.*

3) "Deuterated butanol" is abbreviated as "d-butanol". Neither of these terms tell the reader what the molecule actually is, and more clarification is required. Indeed, the prefix "d-" (as opposed to "l-") usually specifies the rotation direction of optical activity...and this is not relevant in this case. Assuming "deuterated butanol" is the same species for which Barmet et al. (2012) measured the OH rate coefficient(line 139), it is CD3CD2CD2CD2OH. This should be identified as "1-butan-d9-ol" or"butanol-d9".

*Thank you for the clarification on butanol-d9. The molecule is the same as described by Barmet et al. We've updated the molecular name to be consistent with your recommendation for butanol-d9*

4) A little more explanation could be provided on the deviations between chamber and ambient chemistry that occur at high OHR, and why values of 70 s-1 and below are acceptable.

*Peng et al. (2016) recommend OHR values less than ~50 - 100 $s^{-1}$ when operating oxidation flow reactors using 254 nm. We now note this in the text.*

5) The use of the MCM a-pinene mechanism alone to represent all monoterpenes seems unnecessarily simplistic. As stated, the MCM includes a-pinene, b-pinene and limonene as examples of monoterpenes respectively containing endocyclic- ,exocyclic- and endocyclic- + exocyclic- double bonds, and therefore has significant coverage of the list given previously on line 390. An improved approach would surely have been to represent those three monoterpenes explicitly, and use them as mechanistic surrogates for the remaining ones. For example, the OH- and O3- initiation reactions for camphene (containing an exocyclic- double bond) could be included explicitly in the mechanism, but represented to form the corresponding products in the b-pinene scheme (this being the closest structural representative).

*We agree with the reviewer that this may be an over simplification. The purpose of Figure S3 is primarily to show that the signal we detect at m/z 137 is unaffected by the fragmentation of other masses detected by PTR-ToF-MS. We do this because the monoterpenes are a significant contributor to OH reactivity in the mini chamber, and we want to be sure that are only quantifying monoterpenes.*

*We originally treated all monoterpenes as alpha pinene simply to capture the primary OH reactivity; however, we do agree that some degree of speciation is pertinent. In Figure S3, we now also include the monoterpene speciation for*

*Engelmann spruce. We feel the best approach would be to use this speciation for the F26 experiment (Figure S3), and the speciation reported from Hatch et al. (2015) for ponderosa pine in modeling F38. We now separate the signal measured by PTR-ToF-MS to alpha pinene, beta pinene, and limonene using the monoterpene speciation described above, then lump other species to these surrogates based on exocyclic- and endocyclic- double bonds. We have added these details to the main text.*

6) Hydroperoxide photolysis is a slow process in the atmosphere, even if much more rapid with 254 nm illumination in the chamber. What photolysis rates are used? Reaction with OH will almost certainly dominate in the atmosphere. Have you investigated the effect of assuming a radical forming process (photolysis) in place of a radical propagating process (OH reaction) in the ambient plume simulations?

*We apologize, our explanation of the hydroperoxide fate was incomplete and we appreciate this comment. We glossed over these details because in the ambient plume, the fate of the RO2 radical is predominantly reaction with NO; therefore, hydroperoxide formation from RO2 + HO2 is not considered significant for ambient analysis. We recognize that this is an oversight for the mini chamber, and that additional details are needed. We do incorporate OH reactions for the hydroperoxides, and the fate is different depending on structure. For all hydroperoxides, we assume a photolysis frequency equivalent to that of methyl hydroperoxide (J<41> in the MCM). For hydroperoxides with a hydrogen alpha to the peroxide, we assume that this hydrogen is abstracted, and that the resulting radical quickly decomposes to form a carbonyl + OH (example: HYDFURANOOH, Fig S4), which is analogous to the reactions that occur for aromatic and alkane hydroperoxides. For other hydroperoxides (e.g. HYDDIMEFURANOOH, which has no alpha H, Fig S), we assume that the hydrogen of the peroxide group is abstracted, and that the RO2 radical is recycled. All of these reactions occur with a rate constant of 4e-11 (consistent with other hydroperoxides in the MCM). We did not list these reactions in the supplemental figures, but have now added these for clarity. We have also clarified this in the main text.*

7) It would be a very helpful service to the community if the added chemistry was also provided in the form of a reaction listing that can be used in conjunction with MCM v3.3.1. It should also be made clear that expressions like "the updated MCM" used at various points in the manuscript do not imply that the MCM itself has been updated. Perhaps "the extended MCM" or "the customized MCM" would be more appropriate.

*We agree with the reviewer on phrasing and have changed "updated MCM" to "customized MCM". We also agree that including the mechanism would be useful to the community, and we have added the FACISIMILE file as supplemental information. This file contains all of the reactions used to model the mini chamber*

*and ambient plume, but a readme file explains which reactions were added based on this work.*

8) kRO2HO2 actually has a size-dependence term in the MCM, based on carbon number (see Saunders et al., 2003). The quoted rate constant is the limiting value at high carbon number. Was this taken into account?

   *Thank you for catching this. We did not incorporate the size-dependence term in our previous draft. We now include these factors (0.62 for C4, 0.7 for C5 RO2s, 0.77 for C6), which is reflected in the main text, mechanisms, and SI figures.*

9) Although MCM v3.3.1 does not generally contain RO2 H-shift isomerization reactions, it does contain those specifically identified to occur during the degradation of isoprene (see Jenkin et al., 2015), which is included in the speciation applied in this study. Perhaps this statement should be qualified accordingly.

   *We've added a statement qualifying the isomerization reactions.*

10) If butanol-d9 is a significant contributor to OH reactivity, this approach would seem over simplified. Fig. 7 suggests use of 45 ppb, equating to and OH reactivity of about 3.8 s-1 - compared with about 2.3 s-1 for 5-methylfurfural and 2-methylfurancollectively. If this latter pair is significant (as stated on line 399) then butanol-d9 is even more significant and it would seem prudent to represent its chemistry more explicitly.

   *The statement about OH reactivity for furans is specifically aimed at assessing missing reactivity in biomass burning emissions, rather than the mixture in the mini chamber. We appreciate that butanol-d9 will contribute to OH reactivity in the mini chamber (our estimates are ~5%), but we expect that the products will only be a small fraction of reactivity. We feel that the reaction that we do add for butanol-d9 is enough to capture the primary OH reactivity of this species (butanol-d9 + OH = butanol-d9 products + HO2). Since we are focused on developing a mechanism that could be used for ambient biomass burning plumes, we don't aim to dig deeply into the chemistry of butanol-d9.*

11) There is also the possibility that its degradation might form products of masses that interfere with those from the studied oxygenated compounds, and this should be checked. Based on a quick appraisal, I think one product type (C4H2D8O3 - i.e.CD3CD2CD(OOH)CD2OH and CD3CD(OOH)CD2CD2OH) is formed in the first generation of oxidation of butanol-d9, from RO2 + HO2 reactions. This has an interfering mass of interest (114), which I believe is isobaric with methyl hydroxy furanone and beta-acetylacrylic acid.

*Our PTR-ToF-MS has the resolution (R ~ 4000) to separate isobaric species. In general, we are only concerned with overlaps in isomers. We do detect the products of butanol-d9 oxidation (mostly as proton-transfer fragments); however, these species have a significant positive mass defect that does not interfere with the quantification of non-deuterated molecules.*

12) Presumably, the slower decay might also result from formation of isobaric products in the system?

*This is a very good point. We've added a sentence mentioning that part of this disagreement could be due to the formation of isomers.*

13) Statements like "....chemical pathways that are unaccounted for within MCM v 3.3.1" are very easy to include as explanations, but are often not backed up by evidence or specific examples. The degradation of almost all VOCs produces formaldehyde, and this is actually generally found to be substantial and quite well represented in MCM. For example, looking at your list of species, the 3 most abundant (after formaldehyde itself) are 1-butene, methanol and ethene, with these making a collectively significant contribution to the OH reactivity. 1-butene and methanol generate formaldehyde essentially quantitatively (on a molar basis) from radical propagation pathways in the first oxidation step, with ethene producing about 1.7 molecules of formaldehyde. MCM represents this very well, and continues to represent formaldehyde formation from the further oxidation of C2 and higher products (e.g. C2H5CHO from 1-butene). Going on down the list, most of the species make formaldehyde from first- and/or higher-generation chemistry in MCM v3.3.1 - but with the obvious exceptions being species that do not contain a -CH2- substructure (e.g. ethyne, formic acid, glyoxal, phenol and benzene) which cannot make formaldehyde. The much more likely explanation for the under-representation of formaldehyde formation is therefore missing species in the starting speciation. It might also be very sensitive to uncertainties in the [NO] initialization in the model, as this would influence the relative importance of radical propagation and radical termination processes.

*We completely agree with the reviewer, and we appreciate the need for clarifying these semantics. Our intention with that statement was much more broad. When we referred to "chemical processes," we also mean to say that there are likely other precursors of formaldehyde that are not measured well by PTR-ToF-MS, and therefore their chemistry is not represented. There are also uncertainties about photolysis and to what degree does the use of 254 nm light contribute to formaldehyde formation. We've clarified this message to highlight all these uncertainties.*

*We would like to note that the presentation of formaldehyde is primarily to show species that are typically regarded as secondary. We've removed the statement about "Additional work is needed to better describe small oxygenate formation"*

*since these oxygenates are not a primary focus of this study and, as the reviewer points out, this statement is not supported by a thorough analysis.*

14) MCM only includes RO2 + NO2 for acyl peroxy radicals and CH3O2. The reaction is not represented in most cases because the RO2NO2 products are thermally unstable and rapidly regenerate RO2 and NO2 (as indeed the CH3O2 + NO2reaction does at 298 K). The RO2 + NO2 contribution is therefore probably dominated by a very limited set of reactions. The C1 reaction is only CH3O2+NO2, C2 is probably dominated by CH3C(O)O2 + NO2, C3 by C2H5C(O)O2 + NO2: and the small C7contribution is probably mainly C6H5C(O)O2 + NO2. Figure 9 therefore suggests that this limited set of reactions has a big effect. It might also be more correct to check the back decomposition rates for RO2NO2 = RO2 + NO2 and to present the net (forward minus reverse) effect of the RO2 + NO2 reaction - particularly for CH3O2 + NO2 =CH3O2NO2. Note also that the decomposition rates are very sensitive to temperature. Was temperature measured in the mini-chamber and taken into account in the simulations?

*We thank the reviewer for this comment - we agree, this graph is a bit misleading as presented. Our intention with this graph is not to show the net products, but rather the relative importance of each RO2 pathway to a given RO2 species. We agree with the reviewer that this graph inflates the influence of CH3O2NO2 due to the cycling between the RO2 radical and the semi-stable intermediate. As presented, it would seem that NOx losses due to CH3O2NO2 formation are enormous! We do constrain the rate constant based on the temperature measured in the mini chamber, and CH3O2NO2 does fall apart quickly following formation.*

*We've changed this graph to show the relative contributions within each carbon category. We believe this more clearly shows our intended message, that RO2 pathways in ambient plumes and those in the mini chamber were different.*

15) Line 399: "2-dimethylfuran" should be either "2-methylfuran" or "2,5-dimethylfuran".

*We have changed this to read 2,5-dimethylfuran*

16) Line 586: "undesireable" should be "undesirable"

*We have made the correction*

**Reviewer 2 Comments**

Coggon et al., have used an extensive data set of PTR-ToF-MS and I-CIMS measurements, carried out during a wide range of chamber biomass burning /OH

oxidation experiments in order to investigate the impact of a range of observed emitted furans and oxygenated aromatics on the formation of secondary reactive VOC and ozone in aged smoke. This was inferred from model/measurement comparisons of detailed chemical box modelling, where schemes from the Master Chemical Mechanism (MCMv3.3.1)were updated and extended. The results highlight significant formation of a range of C4 and C5 anhydrides and furanone species in aged biomass burning plumes. The derived mechanisms were put into a Lagrangian box model in order to model the chemical evolution of a real life biomass burning again highlighted the importance of furan chemistry and its impact on ozone formation. The work presented is scientifically and mechanistically interesting and the comprehensive compositional measurement dataset, mechanism development and model interpretation work significantly adds to our further understanding of the atmosphereic impact of biomass burning on fundamental atmospheric chemistry and air quality. The work is very much suitable for publication in ACP, after the following comments and clarifications have been addressed.

*We thank the reviewer for their time in reviewing our work and helpful comments. Please find our responses below.*

1) "mini-chamber" experiments and the use of UVC lamps as a large, "clean" source of OH

   The burn smoke is diluted with humidified air (why 30%?) and introduced into the mini-chamber along with ozone (why 70 ppb?), where the chamber reactive mixture is exposed to (predominantly?) 254 nm UVC radiation in order to produce OH to initiate the chemical degradation of the chamber mix. Therefore, the chemistry in the system is not carried out under atmospherically relevant conditions, these studies are aimed mainly at looking at the mechanisms of the OH chemistry. The chemistry is dominated by OH chemistry, with little information given on the NOx and NOy composition in the mini-chamber (only estimated from stack measurements) and any photolysis chemistry of photo-labile species is occurring at non-atmospherically relevant wavelengths.

   *We thank the reviewer for this comment. Before addressing each point specifically, we wish to direct the reviewer to our summary of revisions. In this updated manuscript, we explicitly outline the scope of the mini chamber modeling (i.e., identifying the secondary NMOG detected by PTR-ToF-MS and I-CIMS) and provide more details about the estimations of photolysis frequencies. In our previous draft, we discussed the uncertainties associated with mini chamber UVC lights, and we quantified the extent to which non-OH losses were responsible for the evolution of primary and secondary NMOG (Sections 3.2.2). We also compared the chemistry of the mini chamber with that expected for an ambient biomass burning plume in order to benchmark mini chamber observations (Section 3.2.3).*

*Smoke was diluted with humid air (RH = 30%) and mixed with 70 ppb of ozone in order to simulate 1 -5 days of atmospheric oxidation over the course of 40 minutes. We have added text to note this explicitly in Section 2.2*

*NOx and NOy measurements were not available in the mini chamber. Our previous draft discusses our estimations of NO, NO2, and HONO initial conditions at lines 196-197. We have moved this up to Section 2.2 for clarity.*

2) There is little information given on the chamber auxiliary chemistry mechanism needed in order to the background reactivity of the chamber in the chamber specific box modelling. For example, processes that need to be taken into account include: introduction of free radicals from heterogeneous chamber wall reactions; adsorption/desorption of NOy species (including HONO) to/from the chamber walls; off-gassing of various reactive species from the chamber walls, which can contribute significantly to the radical budget of the system. Also, initial HONO concentration is important to know. Is a detailed chamber auxiliary mechanism available and used here? Formic and acetic acid are shown to be significant secondary product VOCs in the experiments (Line 288).These small oxygenates, as well as HCHO and HONO have significant wall sources in Teflon chambers. Have these sources been taken into account? (in the chamber auxiliary mechanism).

*As mentioned in our summary of revisions, our intention in modeling the mini chamber is to understand the identities and processes contributing to the formation of NMOG measured by PTR-ToF-MS and I-CIMS. Therefore, we focus strictly on gas-phase oxidation processes leading to the formation of maleic anhydride and hydroxy furanones (i.e., furan oxidation). We agree that chamber surfaces could affect our measurements through gas-wall partitioning; however, the OH environment in the chamber is overwhelmingly controlled by ozone photolysis. Likewise, an auxiliary mechanism would be complicated by the mini chamber operation since the Teflon bag is continuously diluted with clean, humid air. Biomass burning emissions are notoriously semi-volatile (Bian et al. 2015, Hodshire et al. 2019), and accounting for wall-vapor interactions would be a modeling effort in itself and would mostly be useful for understanding SOA formation. Despite this impact, Lim et al. (2019) show that the primary VOCs measured in the mini chamber did not exhibit significant wall losses prior to photochemistry.*

*We agree with the reviewer that it is important to note that we are not considering these processes in the model, and have noted this in Section 2.4.*

3) It is stated at line 238/239 that "radical reactions and NOx loss process are likely sensitive to initial NO/NO2 ratios", have any model sensitivity analyses been done to look at this sensitivity?

*We have varied the NO/NO2 ratio - changes in this ratio do not impact OH concentrations, and any NO is quickly converted to NO2 within seconds of initiating the model. The larger effect is mostly in estimating absolute NO concentrations, which impacts the initial RO2 pathways shortly after initiating the model. Higher NO leads to greater RO2 + NO, which can lead to sharp increases in first-generation NMOG, which are not observed in I-CIMS data. Since we do not have NOx measurements, our best assumption of the NO/NO2 ratio is based on modeling the evolution of NO + NO2 in the mini chamber, prior to photochemistry. Within the stated distribution (> 95% NO2), we do not see significant impacts on mini chamber results.*

4) More chamber details are needed – mixing time, spectrum of the lamp used and how was the photon flux derived; how uniform throughout the chamber is the photon flux exposure?

   *The mixing processes are captured by the evolution of acetonitrile, which is represented in the modeling. We also discuss the mixing procedure in Section 2.2. The UVC lights emit at a narrow wavelength at 254 nm (line 106). We cannot comment on the uniformity of the photon flux exposure; however, the Teflon chamber was enclosed in a reflective box. Therefore, we can only assume a uniform photon distribution.*

5) As the other reviewer points out, information on the photolysis rates of some of the primary photo-labile VOCs, along with rate constants for other atmospherically important oxidants are given in the SI. However, little to no information has been given as to the sources of these important data are given. Also, how were the photolysis rates calculated? It is important to note that the MCM chemical mechanisms are designed for use in tropospheric chemistry models, where the cross-sections and quantum yield data used to derive the photolysis rates are mainly used at > 290 nm. Therefore, any primary and secondary MCM chemistry used here in the chamber modelling (but not in the BB plume models) needs to be adjusted to photolysis < 290 nm (mainly around254 nm). For example, the 1,4-dicarbonyl chemistry (mainly 2-butenedial and 4-oxo-2-pentenal, which are significant products from the atmospheric chemistry of aromatics and furans) in the MCM is mainly driven by photolysis (see Newland et al., 2019),where the main photolysis products are furanones and maleic anhydride (a main focus of this study). However, at 254 nm, the photolysis product distribution (quantum yields and photo-tautomer distributions) are different (Tang and Zhu 2005).

   *We appreciate the reviewers careful critique of our discussion about photolysis. We refer the reviewer to our summary of revisions. We have included new tables into the SI that describe our assumptions about the photolysis frequencies used for mini chamber modeling.*

*We note that our mechanism development is limited towards including reactions relevant to ambient biomass burning plumes since our intention is to use this mechanism to understand ozone and secondary NMOG formation in ambient smoke (Section 3.2.4). We do appreciate that some of the important intermediates (e.g. butenedial) could have different photolysis pathways at 254 nm. However, when we estimate the photolysis frequency of butenedial at 254 nm (which we note is not well studied; the cross sections dramatically decrease between 230 - 260 nm and previous studies have investigated absorption at these wavelengths with limited resolution), we find that photolysis only contributes ~ 5% to the total loss of butenedial. In the mini chamber, OH concentrations are greatly elevated and OH + butenedial is the predominant loss pathway.*

6) I agree with the other referee that given that a) monoterpenes contribute significantly to the calculated OH reactivity on the experiments and b) the MCM contains extensive chemical mechanism for the atmospheric degradation of both alpha and beta pinene,as well as limonene (and beta caryophellene), using a-pinene as a proxy is an over-simplification here (Line 394).

*We agree with both reviewers and have separated PTR-ToF-MS measurements to alpha pinene, beta pinene, and limonene to account for differences in reactivity associated with endocyclic and exocyclic double bonds. Please see comment (5) in our response to Reviewer 1, and also new text in Section 3.2.1.*

7) Line 80 – define "0-D"

*We removed "0-D" since "box model" is descriptive enough.*

8) Line 105 – define "semi-batch"

*We have removed "semi-batch" as a descriptor since we provide details for chamber operation.*

9) Line 275 – why is an OH atmospheric concentration of 1.5E+6 cm-3 assumed?

*This atmospheric concentration is commonly used in OFR studies (Peng 2016, 2018).*

10) Line 353 – "I-ToF-MS" should be I-CIMS?

*We have corrected I-ToF*

11) Line484 – reference needed for the o-semiquinone chemistry

*We have included a reference to Finewax et al., 2018.*

12) Line 501 – is this exhibiting a "bi-exponential" decay profile?

*It is not clear to what degree this is exhibiting a bi-exponential decay. We argue that there is likely formation of a secondary species that is isomeric with furfural (Lines 505-510).*

13) Line 515 – could the model be underpredicting the MA yield owing to increased MA and furanone yields from 1,4-dicarbonyl photolysis at 254 nm?

*This is certainly possible; however, it seems unlikely because 1,4-dicarbonyl formation is not immediate, as is MA formation at the beginning of the experiment. We also note that the modeled losses of 1,4-dicarbonyls seem to be mostly due to OH rather than photolysis (see comment 5).*

14) Line 537 and Figure 2 – under-prediction of small oxygenates – some of this could be coming from wall sources? (e.g. Zador et al., 2005)

*We think that is also very likely. We lump wall sources into the "heterogeneous" processes. We've clarified that this could be aerosol or wall heterogeneous chemistry.*

15) Line 560 – ozonolysis of other monoterpenes than a-pinene could be more important?

*We have updated the model to include a more detailed speciation of monoterpenes (see comment 6). Most of the monoterpenes are still represented by alpha-pinene, and the fate by ozonolysis remains ~1%. We have updated the text to indicate that we are discussing the sum of monoterpenes.*

16) Line 718 – "understudied NMOG OH chemistry" – this study really focuses on the OH chemistry

*Thank you for catching this - we've qualified that this is OH chemistry.*

17) Line 721 – what does "understory" mean?

*Understory refers to a burn where smaller biomass beneath the canopy are burned (e.g. twigs, leaves, needles, etc).*

18) Figure S6, S7 and S9 – J<41> (CH3OOH photolysis rate) – has this been adapted for 254 nm photolysis?

*J<41> has been adapted for 254 nm photolysis (now clarified in text, and in Table S1).*

19) Figure S10 – what is GUAIACOLPROD?

*GUAIACOLPROD is a generic, non-reactive product that is included to account for the fraction of guaiacol that does not react to form nitroguaiacols. We note this now in the figure caption.*

20) Figure S14 – give structure of HYDMEFURANO2 on plot.

*Thank you for the suggestion - we have added HYDMEFURANO2 and HYDFURANO2 structures to the plot.*

**Reviewer 3 Comments**

This paper provides some important insights into emissions from biomass burning. Biomaterial was burned and chemical measurements of the resulting emissions. The MCM chemical mechanism was used in simulations of the experiments. The authors added more chemical reactions to the MCM to improve its performance in simulating the experiments.

Much more data like this is needed for the continuing development chemical mechanisms for atmospheric chemistry modeling. Also the authors have used some quantum chemistry in the further development of the test reactions that were added to the MCM. Finally, they considered real biomass burning plumes.

All in all, this paper represents a good mix of laboratory measurements, mechanism development with evaluation and applications to the real world. It is an excellent template for future studies.

Minor: Lines 210 - 215 I commend the authors for using the correct term, photolysis frequencies in this section. However, they need to be consistent and state at the beginning: "Photolysis frequencies are calculated using literature cross-sections and quantum yields of relevant organic and inorganic species..."

*We thank Dr. Stockwell for his kind words. We have corrected the terminology here, and at other points in the manuscript.*

[revised manuscript text omitted]

The I-CIMS utilizes a "soft" chemical ionization source that forms iodide clusters with polarizable analyte molecules (Huey et al., 1995; Lee et al., 2014). The instrument used here was operated in a similar configuration to that described in Krechmer et al. (2016). To generate reagent ions, 2 SLPM of clean $N_2$ from dewar blow-off was run over a methyl iodide permeation tube and ionized using a Polonium-210 ionizer and into an ion molecule reaction region (IMR). The I-CIMS measured gas-phase signals from the mini chamber at 1 Hz time resolution. Smoke was diluted with 4 L min$^{-1}$ of clean, humidified air at a 4:1 ratio to minimize reagent ion depletion. A constant flow of isotopically labeled formic acid was delivered to the instrument to measure consistency of response. Reported here are I-CIMS measurements normalized to $1\times10^6$ counts per second of the reagent ion signal at m/z 126.905 (normalized counts per second, ncps). Due to unavailability of standards, I-CIMS data are not reported in mixing ratios. Secondary NMOG measured by I-CIMS are assigned identifications based on modeling results and previous literature.

**2.3.2 NO$_x$ Measurements**

NO, $NO_2$, and HONO were measured by an open-path Fourier transform infrared spectrometer (OP-FTIR) as described by Selimovic et al. (2018). The OP-FTIR was located on the platform and sampled smoke across the diameter of the stack. The OP-FTIR provides fast measurements and avoids potential sampling artifacts due to sample line losses. OP-FTIR measurements were used for most experiments. $NO_2$ and HONO were also measured by the NOAA Airborne Cavity Enhanced Spectrometer (ACES) as described by Zarzana et al. (2018). ACES was located on the platform and sampled smoke through a 1 m Teflon inlet. These data were used when OP-FTIR data were unavailable, or when $NO_x$ emissions were below OP-FTIR detection limits. Supplementary NO measurements were provided by a custom-built chemiluminescence instrument located in a room on the burn chamber floor. A full description of that instrument is provided by (Stockwell et al., 2018).

**2.4 Box Model Implementation and Evaluation**

NMOG oxidation processes were simulated using the Framework for 0-D Atmospheric Modeling (F0AM v 3.1, https://sites.google.com/site/wolfegm/models, Wolfe et al., 2016). A modified NMOG oxidation mechanism is applied based on the Master Chemical Mechanism (MCM v. 3.3.1, Jenkin et al., 1997, 2003, 2015; Saunders et al., 2003). As described below and in Section 3.2.1, the box model is used to evaluate NMOG formation in the mini chamber, as well a for the daytime oxidation of an ambient biomass burning plume (Müller et al., 2016). Non-atmospheric photolysis played a role in the evolution of organics in the mini chamber owing to the use of UVC lamps; consequently, interpretation of the mini chamber measurements is limited

to evaluating the link between OH oxidation of furan and oxygenated aromatics with the formation of several key secondary NMOG measured by I-CIMS and PTR-ToF-MS. The ambient model is used to quantitatively evaluate the impact of furan and oxygenated aromatic chemistry on secondary NMOG and ozone formation in real biomass burning plumes.

NMOG chemistry and dilution are assumed to follow the first-order differential equation described by Eqn 1.

$$\quad \frac{dC_i}{dt} = \sum_{m}^{n} P_m r_m - \sum_{m}^{n} R_m r_m - k_i C_i \quad\quad\quad (1)$$

Where $C_i$ is the concentration of species $i$, $P_m$ is the stoichiometric coefficient of reaction $m$ leading to the formation of species $i$, $R_m$ is the stoichiometric coefficient of reaction $m$ leading to the loss of species $i$, $r$ is the elementary rate of reaction $m$, and $k_i$ is a first-order dilution rate constant. A constant dilution term is applied to all species based on the measured loss rate of acetonitrile. Vapor-wall interactions are not considered in this model, and these processes may impact the evolution of intermediate and semi-volatile gases. Lim et al. (2019) did not observe significant wall losses of primary VOCs during chamber experiments. Vapor-wall interactions have been shown to play a role in the SOA evolution of biomass burning smoke, and partitioning to the gas-phase is greater in diluting systems (Bian et al., 2015; Hodshire et al., 2019).

The MCM treats the photolysis of organic and inorganic species through parameterizations relevant to atmospheric wavelengths. To account for photolysis in the mini chamber, MCM photolysis frequencies were calculated using literature cross-sections and quantum yields of relevant organic and inorganic species (Atkinson et al., 2006; Burkholder et al., 2015; Keller-Rudek et al., 2013). Tables S1 and S2 summarize the reactions and databases used to estimate photolysis frequencies in the mini chamber. For some primary species and $NO_x$ reservoirs (e.g., acetone, acetaldehyde, benzaldehyde, PAN), additional reactions were added to account for photolysis pathways at 254 nm (Table S2).

Photolysis frequencies are calculated as the product of the absorption cross section, quantum yield, and photon flux at 254 nm. The measured 254 nm photon flux ($3\times10^{15}$ photons cm$^{-2}$ s$^{-1}$) is scaled by a factor of 1.5 in order to reproduce the measured OH-loss of butanol-d9. No other changes are applied to the rate constants or reactions of MCM v. 3.3.1. To account for other reactive NMOG, literature mechanisms for furan, 2-methylfuran, 2,5-dimethylfuran, furfural, 5-methylfurfural, and guaiacol are included. A full description of these modifications are provided in Section 3.2.1. Table S3 summarizes the photolysis frequencies and OH, $O_3$, and $NO_3$ rate constants for the primary NMOG species modeled here.

[revised manuscript text omitted]

important for acyl $RO_2$ species (Orlando and Tyndall, 2012; Peng et al., 2019). For $RO_2$ + NO reactions, it is assumed that the alkoxy radical quickly decays (either by thermal degradation or reaction with $O_2$) to form carbonyls. Aschmann et al. (2014) did not report species consistent with alkoxy isomerization; thus, these reactions are ignored. Similarly, $RO_2$ + $RO_2$ reactions are assumed to only form alkoxy radicals, which may subsequently degrade to from carbonyls. Peroxides are assumed to be the only products of $RO_2$ + $HO_2$ reactions. These species are assumed to undergo photolysis to form carbonyls. Peroxides may also react with OH, and the resulting products differ depending on structure. For structures with an alpha hydrogen, it is assumed that OH abstracts at the alpha position, and that the resulting radical quickly decomposes to form a carbonyl + OH (e.g. HYDFURANOOH, Figure S4). For other structures, it is assumed that the hydrogen of the peroxide group is abstracted to regenerate the $RO_2$ radical.

The generic MCM rate constants are applied for $RO_2$ + $HO_2$ and $RO_2$ + NO reactions ($k_{RO2NO}$ = 2.7 $\times 10^{-12}$ exp(360/T) $cm^3$ molec $^{-1}$ s$^{-1}$, $k_{RO2HO2}$ = 2.91 $\times 10^{-13}$ exp(1300/T)[1 – exp(-0.245$n$)] $cm^3$ molec $^{-1}$ s$^{-1}$). The $RO_2$ + $HO_2$ rate constant is adjusted for the $RO_2$ carbon number, $n$, as recommended by Saunders et al. (2003). Photolysis frequencies for peroxides are assumed to be the same as for methyl hydroperoxide ($J_{41}$ in the MCM), and peroxide + OH reactions are assumed to have a rate constant of $4\times10^{-11}$ $cm^3$ molec $^{-1}$ s$^{-1}$. The assumed rate constants for $RO_2$ + $RO_2$ reactions are chosen based on those of structurally similar $RO_2$ radicals reported in the MCM. $RO_2$ H-shift isomerization (autoxidation) is not broadly represented in MCM v. 3.3.1, aside for isoprene oxidation (Jenkin et al., 2015). $RO_2$ isomerization becomes competitive when the bimolecular lifetime of $RO_2$ is on the order of 10s (Crounse et al., 2013; Praske et al., 2018). Based on the modeled concentrations of $HO_2$, NO, and $RO_2$, the bimolecular lifetime of $RO_2$ radicals from furan oxidation 
[revised manuscript text omitted]

**1 Evaluating NOMOG Losses Through Metal Ductwork**

As described in the main text, smoke was passed through metal ductwork before injection to the mini chamber. Deming et al. (2019) showed that surface partitioning is dependent on NMOG volatility, functionality, and displacement processes at adsorption sites. Prior to each small chamber experiment, the PTR-ToF-MS sampled NMOG directly from the stack, as described by Koss et al. (2018). To evaluate biases associated with NMOG transmission through the ductwork, we compare the distribution of NMOG measured from the stack with that inside the small chamber prior to the initiation of OH chemistry. Figure S15 shows the difference in NMOG profiles between small chamber and stack measurements (Small Chamber Bias = $[NMOG_i/CH_3CN]_{chamber}$ - $[NMOG_i/CH_3CN]_{stack}$). We normalize the NMOG distribution to $CH_3CN$ under the assumption the acetonitrile is not lost to surfaces to a greater degree than other NMOG. The top row shows bias histograms for three fires, while the bottom row shows a comparison between the normalized profiles with 1:1, 1:2, and 2:1 lines.

In general, most NMOG fall within 20% of the 1:1 line. Bias histograms (top row, Fig. S15) show that the NMOG/$CH_3CN$ ratios are lower in the small chamber, which suggests that NMOG are lost to the ductwork; however, this loss appears to be normally distributed and not weighted towards any specific NMOG functionality. Some masses exhibit significantly higher ratios with acetonitrile inside the small chamber (e.g. butenes, ethanol, formamide for F29); however, PTR-ToF-MS detection of these masses is poor due to contributions of fragments from higher masses (e.g. butene), or low sensitivity (e.g. ethanol).

Fig. S15 demonstrates that the relative NMOG distribution in the chamber is not significantly different from the NMOG distribution sampled from the stack. These results are consistent with the conclusions drawn by Lim et al. (2019), which showed that the volatility distribution was not significantly different between stack and mini chamber measurements.

**2   Sensitivity of Modeled Secondary NMOG Formation to Furfural Branching Ratios**

The reactions employed to represent furfural oxidation were estimated by Zhao and Wang (2017) via theoretical quantum chemistry calculations. To date, this mechanism has not been studied experimentally; consequently, the exact branching ratios of the three major pathways may differ from those used in this study (0.37 for channel A, 0.6 for channel B, and 0.03 for channel C, 6). Furfural plays a major role in the formation of secondary NMOG measured in the biomass burning plume described by Müller et al. (2016) and the assumed branching ratios may impact modeled formation of maleic anhydride, hydroxy furanone, and ozone.

Figure S16 shows model output of maleic anhydride, hydroxy furanone, and ozone for base case (A = 0.37, B = 0.6, B = 0.03), equal weight (0.33, 0.33, 0.33), and isolated channel (i.e., all channel A, B, or C) simulations of the biomass burning plume described by Müller et al. (2016). Overall, hydroxy furanone formation is most sensitive to the assumed branching ratio of channel B, which is the pathway that directly leads to hydroxy furanone formation (Fig. 6). Maleic anhydride is most sensitive to the assumed branching ratio of channel C; however, this sensitivity is weaker than that of hydroxy furanone since all pathways lead to a significant yield of maleic anhydride. The assumed branching ratios have little impact on ozone formation.

The sensitivity tests presented in Fig. S16 demonstrate the need for experimental evaluation of the furfural oxidation mechanism. This refinement may provide better constraints of important secondary NMOG; however, this will unlikely affect modeled ozone formation.

**Table S1.** References and calculated photolysis frequencies for reactions used in the MCM v 3.3.1 during mini-chamber experiments. Photolysis frequencies are calculated as the product of the absorption cross section, quantum yield, and the scaled photon flux at 254 nm ($4.5 \times 10^{15}$ photons cm$^2$). The ratio to $j_{NO2}$ is shown for ease of comparison.

| Absorbing Species | MCM Reaction | MCM Name | Value (s$^{-1}$) | j / j$_{NO2}$ | Cross Section Database | Quantum Yield Ref. | QY (254 nm) |
|---|---|---|---|---|---|---|---|
| ozone | O3 → O1D | J1 | 4.5E-02 | 885.38 | JPL | JPL | 0.9 |
| ozone | O3 → O | J2 | 5.0E-03 | 98.38 | JPL | JPL | 0.1 |
| hydrogen peroxide | H2O2 → OH + OH | J3 | 3.0E-04 | 6.03 | JPL | JPL | 1 |
| nitrogen dioxide | NO2 → NO + O | J4 | 5.0E-05 | 1.00 | IUPAC | IUPAC | 1 |
| nitrate radical | NO3 → NO2 + O | J5 | 4.9E-04 | 9.79 | Mainz | JPL | 1 |
| nitrate radical | NO3 → NO | J6 | 0 | 0 | Mainz | JPL | 0 |
| nitrous acid | HONO → OH + NO | J7 | 5.9E-04 | 11.70 | IUPAC | IUPAC | 1 |
| nitric acid | HNO3 → OH + NO2 | J8 | 8.5E-05 | 1.68 | JPL | IUPAC | 1 |
| formaldehyde | HCHO → CO + HO2 + HO2 | J11 | 4.5E-06 | 0.09 | JPL | JPL | 0.3 |
| formaldehyde | HCHO → H2 + CO | J12 | 7.5E-06 | 0.15 | IUPAC | JPL | 0.5 |
| acetaldehyde[1] | CH3CHO → CH3O2 + HO2 + CO | J13 | 2.0E-05 | 0.39 | IUPAC | IUPAC | 0.3 |
| propanal | C2H5CHO → C2H5O2 + HO2 + CO | J14 | 7.6E-05 | 1.50 | IUPAC | IUPAC | 1 |
| butanal[2] | C3H7CHO → NC3H7O2 + CO + HO2 | J15 | 1.3E-05 | 0.26 | IUPAC | IUPAC | 0.21 |
| butanal | C3H7CHO → C2H4 +CH3CHO | J16 | 6.3E-06 | 0.12 | IUPAC | IUPAC | 0.1 |
| 2-methylpropanal | IPRCHO → IC3H7O2 + HO2 + CO | J17 | 1.4E-05 | 0.27 | IUPAC | IUPAC | 0.25 |
| methacrolein[2] | MACR → CH3C2H2O2 + CO + HO2 | J18 | 3.9E-07 | 0.01 | IUPAC | JPL | 0.05 |
| methacrolein | MACR → MACO3 + HO2 | J19 | 3.9E-07 | 0.01 | IUPAC | JPL | 0.05 |
| (C5)-HPALD | C5HPALD1 → products | J20 | 7.7E-06 | 0.15 | IUPAC | Wolfe et al. (2012) | 1 |
|  | C5HPALD2 → products |  |  |  |  |  |  |
| acetone[1] | CH3COCH3 → CH3CO3 + CH3O2 | J21 | 5.6E-05 | 1.11 | IUPAC | IUPAC | 0.42 |
| methyl ethyl ketone | MEK → CH3CO3 + C2H5O2 | J22 | 4.6E-05 | 0.90 | IUPAC | IUPAC | 0.34 |
| methyl vinyl ketone | MVK → C3H6 + CO | J23 | 4.0E-06 | 0.08 | IUPAC | IUPAC | 0.38 |
| methyl vinyl ketone | MVK → CH3CO3 + HCHO + CO + HO2 | J24 | 4.0E-06 | 0.08 | IUPAC | IUPAC | 0.38 |
| glyoxal | GLYOX → CO + CO + H2 | J31 | 3.6E-05 | 0.71 | JPL | JPL | 0.52 |
| glyoxal | GLYOX → HCHO + CO | J32 | 2.3E-05 | 0.45 | JPL | JPL | 0.33 |
| glyoxal | GLYOX → CO + CO + HO2 + HO2 | J33 | 1.1E-05 | 0.21 | JPL | JPL | 0.16 |
| methyl glyoxal | MGLYOX → CH3CO3 + CO + HO2 | J34 | 1.2E-04 | 2.37 | JPL | JPL | 1 |
| biacetyl | BIACET → CH3CO3 + CH3CO3 | J35 | 2.6E-05 | 0.51 | IUPAC | IUPAC | 0.158 |
| methyl hydroperoxide | CH3OOH → CH3O + OH | J41 | 1.5E-04 | 2.90 | IUPAC | IUPAC | 1 |
| methyl nitrate | CH3NO3 → CH3O + NO2 | J51 | 1.5E-04 | 2.88 | IUPAC | IUPAC | 1 |
| ethyl nitrate | C2H5NO3 → C2H5O + NO2 | J52 | 1.8E-04 | 3.59 | IUPAC | IUPAC | 1 |
| propyl nitrate | NC3H7NO3 → NC3H7O + NO2 | J53 | 1.9E-04 | 3.85 | IUPAC | IUPAC | 1 |
| isopropyl nitrate | IC3H7NO3 → IC3H7O + NO2 | J54 | 2.2E-04 | 4.30 | IUPAC | IUPAC | 1 |

JPL - Burkholder et al. (2015) (https://jpldataeval.jpl.nasa.gov/)

IUPAC - Atkinson et al. (2006) (http://iupac.pole-ether.fr/index.html)

Mainz - Keller-Rudek et al. (2013) (www.uv-vis-spectral-atlas-mainz.org)

[1] Additional photolysis rections added to account for losses at 254 nm (see Table S2)

[2] No Data for QY available at 254 nm; assume highest value reported at higher wavelengths

**Table S2.** Same as Table S1, but for reactions that are likely to be important in the mini-chamber that are not represented in MCM v 3.3.1.

| Absorbing Species | MCM Reaction | MCM Name | Value (s$^{-1}$) | j / j$_{NO2}$ | Cross Section Database | Quantum Yield Ref. | QY (254 nm) |
|---|---|---|---|---|---|---|---|
| benzaldehyde | BENZAL → HO2 + C6H5CO3 | Jn2 | 1.1E-03 | 21.30 | IUPAC | IUPAC | 0.3 |
| | BENZAL → HO2 + CO + C6H5O2 | | | | | | |
| acetaldehyde | CH3CHO → CH4 + CO | Jn5 | 3.0E-05 | 0.60 | IUPAC | IUPAC | 0.46 |
| acetaldehyde | CH3CHO → CH3CO3 + HO2 | Jn6 | 7.9E-06 | 0.16 | IUPAC | IUPAC | 0.12 |
| 2-furfural | FURFURAL → FURAN + CO | Jn7 | 1.3E-01 | 2646.26 | Mainz | Estimate[1] | 0.6 |
| | FURFURAL → C3H4 + CO + CO | | | | | | |
| acetone | CH3COCH3 → CH3O2 + CH3O2 + CO | Jn8 | 2.7E-05 | 0.53 | IUPAC | IUPAC | 0.2 |
| peroxyacetyl nitrate | PAN → CH3CO3 + NO2 | Jn14 | 3.2E-04 | 6.41 | IUPAC | IUPAC | 0.745 |
| peroxyacetyl nitrate | PAN → CH3O2 + NO3 | Jn15 | 1.1E-04 | 2.23 | IUPAC | IUPAC | 0.26 |
| methoxy nitrate | CH3O2NO2 → CH3O2 + NO2 | Jn16 | 1.4E-03 | 27.26 | IUPAC | Estimate | 0.95 |
| methoxy nitrate | CH3O2NO2 → CH3O + NO3 | Jn17 | 7.2E-05 | 1.43 | IUPAC | Estimate | 0.05 |
| dinitrogen pentoxide | N2O5 → NO3 + NO2 | Jn19 | 2.8E-04 | 5.58 | IUPAC | IUPAC | 0.2 |
| dinitrogen pentoxide | N2O5 → NO3 + NO + O | Jn20 | 8.8E-04 | 17.51 | IUPAC | IUPAC | 0.6 |
| HO2NO2 | HO2NO2 → HO2 + NO2 | Jn21 | 9.2E-04 | 18.32 | IUPAC | IUPAC | 0.59 |
| HO2NO2 | HO2NO2 → OH + NO3 | Jn22 | 6.4E-04 | 12.73 | IUPAC | IUPAC | 0.41 |

JPL - Burkholder et al. (2015) (https://jpldataeval.jpl.nasa.gov/)

IUPAC - Atkinson et al. (2006) (http://iupac.pole-ether.fr/index.html)

Mainz - Keller-Rudek et al. (2013) (www.uv-vis-spectral-atlas-mainz.org)

[1] Personal communication with V. Papadimitriou

**Table S3.** Rate constant parameters, photolysis frequency, and initial conditions for the species modeled in F26 and F38. Entries are ordered by mixing ratios measured during F38. Photolysis frequencies are calculated based on literature cross-sections, known/estimated quantum yields, and the scaled photon flux at 254 nm ($4.5 \times 10^{15}$ photons cm$^2$).

| | MCM Name | $k_{OH}$ (cm$^3$ molec$^{-1}$ s$^{-1}$) | $k_{O3}$ (cm$^3$ molec$^{-1}$ s$^{-1}$) | $k_{NO3}$ (cm$^3$ molec$^{-1}$ s$^{-1}$) | $j$ (s$^{-1}$) | F26 (ppb) | F38 (ppb) |
|---|---|---|---|---|---|---|---|
| **Butanol-d9** | DBUTANOL | 3.40E-12 | 0 | 0 | 0 | 37.26 | 44.86 |
| **Formaldehyde** | HCHO | 8.47E-12 | 0 | 5.50E-16 | 1.20E-05 | 14.07 | 17.84 |
| **1-butene** | BUT1ENE | 3.11E-11 | 1.06E-17 | 1.35E-14 | 0 | 4.53 | 13.84 |
| **Methanol** | CH3OH | 9.02E-13 | 0 | 0 | 0 | 22.23 | 11.79 |
| **Ethene** | C2H4 | 7.74E-12 | 1.68E-18 | 2.24E-16 | 0 | 9.22 | 8.38 |
| **Acetaldehyde** | CH3CHO | 1.48E-11 | 0 | 2.84E-15 | 1.98E-05 | 15.44 | 7.86 |
| **Acetic Acid** | CH3CO2H | 8.00E-13 | 0 | 0 | 0 | 7.64 | 6.03 |
| **Ethyne** | C2H2 | 7.46E-13 | 0 | 0 | 0 | 4.97 | 5.96 |
| **Formic Acid** | HCOOH | 4.50E-13 | 0 | 0 | 0 | 3.89 | 4.86 |
| **Acrolein** | ACR | 2.00E-11 | 2.90E-19 | 3.26E-15 | 3.86E-07 | 3.24 | 4.43 |
| **1-propene** | C3H6 | 2.83E-11 | 1.04E-17 | 9.79E-15 | 0 | 7.52 | 2.64 |
| **2-furfural** | FURFURAL | 3.50E-11 | 0 | 0 | 1.34E-01 | 2.07 | 2.03 |
| **Acetone** | CH3COCH3 | 1.78E-13 | 0 | 0 | 8.3E-05 | 5.60 | 1.95 |
| **Furan** | FURAN | 4.20E-11 | 0 | 0 | 0 | 2.26 | 1.87 |
| **Furanone** | BZFUONE | 4.45E-11 | 2.20E-19 | 3.00E-13 | 0 | 1.77 | 1.73 |
| **2,3-butanedione** | BIACET | 2.41E-13 | 0 | 0 | 2.55E-05 | 2.09 | 1.66 |
| **1,3-butadiene** | C4H6 | 6.59E-11 | 6.64E-18 | 1.03E-13 | 0 | 2.40 | 1.42 |
| **Ethanol** | C2H5OH | 3.21E-12 | 0 | 0 | 0 | 1.63 | 1.34 |
| **Glyoxal** | GLYOX | 9.63E-12 | 0 | 2.84E-15 | 6.95E-05 | 0.98 | 1.25 |
| **Hydroxyacetone** | ACETOL | 4.42E-12 | 0 | 0 | 4.55E-05 | 1.30 | 1.12 |
| **Guaiacol** | GUAIACOL | 7.44E-11 | 0 | 0 | 0 | 2.94 | 1.10 |
| **5-methylfurfural** | MEFURFURAL | 5.10E-11 | 0 | 0 | 0 | 1.45 | 1.00 |
| **Catechol** | CATECHOL | 1.00E-10 | 9.20E-18 | 9.90E-11 | 0 | 1.45 | 1.00 |
| **Phenol** | PHENOL | 2.74E-11 | 0 | 3.80E-12 | 0 | 2.39 | 0.89 |
| **Methyl acetate** | METHACET | 3.50E-13 | 0 | 0 | 0 | 1.03 | 0.89 |
| **Propenoic acid** | ACO2H | 8.66E-12 | 0 | 0 | 0 | 0.87 | 0.86 |
| **Methyl vinyl ketone** | MVK | 1.99E-11 | 5.36E-18 | 0 | 8.06E-06 | 0.99 | 0.81 |

| | MCM Name | $k_{OH}$ (cm$^3$ molec$^{-1}$ s$^{-1}$) | $k_{O3}$ (cm$^3$ molec$^{-1}$ s$^{-1}$) | $k_{NO3}$ (cm$^3$ molec$^{-1}$ s$^{-1}$) | j (s$^{-1}$) | F26 (ppb) | F38 (ppb) |
|---|---|---|---|---|---|---|---|
| o-cresol | CRESOL | 4.65E-11 | 0 | 1.40E-11 | 0 | 2.58 | 0.79 |
| alpha-pinene | APINENE | 5.20E-11 | 9.53E-17 | 6.15E-12 | 0 | 0.76 | 0.78 |
| Methyl glyoxal | MGLYOX | 1.29E-11 | 0 | 6.82E-15 | 1.20E-04 | 0.69 | 0.69 |
| Benzene | BENZENE | 1.22E-12 | 0 | 0 | 0 | 1.93 | 0.67 |
| 2-methylfuran | MEFURAN | 6.19E-11 | 0 | 0 | 0 | 1.35 | 0.64 |
| Toluene | TOLUENE | 5.59E-12 | 0 | 0 | 0 | 1.90 | 0.46 |
| Isoprene | C5H8 | 9.91E-11 | 1.33E-17 | 7.03E-13 | 0 | 1.04 | 0.42 |
| Methyl ethyl ketone | MEK | 1.11E-12 | 0 | 0 | 4.55E-05 | 1.47 | 0.41 |
| Acetic Anhydride | METHCOACET | 1.00E-14 | 0 | 0 | 0 | 0.33 | 0.30 |
| p-benzoquinone | PBZQONE | 4.60E-12 | 0 | 3.00E-13 | 0 | 0.24 | 0.30 |
| 2,5-dimethylfuran | DIMEFURAN | 1.32E-10 | 0 | 0 | 0 | 0.75 | 0.28 |
| Methacrolein | MACR | 2.84E-11 | 1.28E-18 | 3.40E-15 | 7.72E-07 | 0.29 | 0.24 |
| Benzaldehyde | BENZAL | 1.25E-11 | 0 | 2.40E-15 | 1.1E-3 | 0.28 | 0.19 |
| 2,3-dimethyl phenol | OXYLOL | 8.00E-11 | 0 | 3.20E-11 | 0 | 0.71 | 0.16 |
| Propenal | C4ALDB | 3.40E-11 | 1.58E-18 | 6.00E-15 | 5.56E-07 | 0.18 | 0.15 |
| Styrene | STYRENE | 5.80E-11 | 1.70E-17 | 1.50E-12 | 0 | 0.36 | 0.11 |
| 1-pentene | PENT1ENE | 3.10E-11 | 1.00E-17 | 1.20E-14 | 0 | 0.34 | 0.10 |
| m-xylene | MXYL | 2.31E-11 | 0 | 2.60E-16 | 0 | 0.26 | 0.09 |
| p-xylene | PXYL | 1.43E-11 | 0 | 5.00E-16 | 0 | 0.26 | 0.09 |
| o-xylene | OXYL | 1.36E-11 | 0 | 4.10E-16 | 0 | 0.17 | 0.06 |
| Ethyl benzene | EBENZ | 7.00E-12 | 0 | 1.20E-16 | 0 | 0.08 | 0.03 |
| NO2 | NO2 | 9.22E-12 | 3.72E-17 | 1.21E-12 | 5.05E-05 | 6.46 | 53.98 |
| O3 | O3 | 7.41E-14 | 4.96E-02 | 0 | 0 | 5.00 | 10.00 |
| HONO | HONO | 5.95E-12 | 0 | 0 | 5.90E-04 | 0.00 | 2.58 |
| NO | NO | 8.97E-12 | 1.78E-14 | 2.60E-11 | 0 | 0.08 | 0.40 |

**Table S4.** Calculated NMOG losses by reaction with OH, $O_3$, $NO_3$, and photolysis for F26, F38, and the ambient biomass burning plume described by Müller et al. (2016). All values are percentages of the integrated loss over 15 hr of atmospheric-equivalent OH oxidation. Entries are ordered according to the largest loss rates by each process, calculated for F38. Entries marked by a hyphen were not included in the modeling.

*Primary Loss by OH*

| | F26 | | | | F38 | | | | Muller et al. (2016) | | | |
|---|---|---|---|---|---|---|---|---|---|---|---|---|
| | OH | $O_3$ | $NO_3$ | *hv* | OH | $O_3$ | $NO_3$ | *hv* | OH | $O_3$ | $NO_3$ | *hv* |
| **5-methylfurfural** | 100 | 0 | 0 | 0 | 100 | 0 | 0 | 0 | 100 | 0 | 0 | 0 |
| **2,5-dimethylfuran** | 100 | 0 | 0 | 0 | 100 | 0 | 0 | 0 | 100 | 0 | 0 | 0 |
| **Guaiacol** | 100 | 0 | 0 | 0 | 100 | 0 | 0 | 0 | — | — | — | — |
| **Furan** | 100 | 0 | 0 | 0 | 100 | 0 | 0 | 0 | 100 | 0 | 0 | 0 |
| **2-methylfuran** | 100 | 0 | 0 | 0 | 100 | 0 | 0 | 0 | 100 | 0 | 0 | 0 |
| **Acetic Acid** | 100 | 0 | 0 | 0 | 100 | 0 | 0 | 0 | 100 | 0 | 0 | 0 |
| **Propenoic acid** | 100 | 0 | 0 | 0 | 100 | 0 | 0 | 0 | — | — | — | — |
| **Acetic Anhydride** | 100 | 0 | 0 | 0 | 100 | 0 | 0 | 0 | — | — | — | — |
| **Methanol** | 100 | 0 | 0 | 0 | 100 | 0 | 0 | 0 | — | — | — | — |
| **Benzene** | 100 | 0 | 0 | 0 | 100 | 0 | 0 | 0 | 100 | 0 | 0 | 0 |
| **Toluene** | 100 | 0 | 0 | 0 | 100 | 0 | 0 | 0 | — | — | — | — |
| **Ethene** | 100 | 0 | 0 | 0 | 100 | 0 | 0 | 0 | 97 | 3 | 0 | 0 |
| **Formic Acid** | 100 | 0 | 0 | 0 | 100 | 0 | 0 | 0 | 100 | 0 | 0 | 0 |
| **Ethanol** | 100 | 0 | 0 | 0 | 100 | 0 | 0 | 0 | — | — | — | — |
| **Methyl acetate** | 100 | 0 | 0 | 0 | 100 | 0 | 0 | 0 | — | — | — | — |
| **m-xylene** | 100 | 0 | 0 | 0 | 100 | 0 | 0 | 0 | — | — | — | — |
| **Ethyl benzene** | 100 | 0 | 0 | 0 | 100 | 0 | 0 | 0 | — | — | — | — |
| **o-xylene** | 100 | 0 | 0 | 0 | 100 | 0 | 0 | 0 | — | — | — | — |
| **p-xylene** | 100 | 0 | 0 | 0 | 100 | 0 | 0 | 0 | — | — | — | — |
| **Acrolein** | 100 | 0 | 0 | 0 | 100 | 0 | 0 | 0 | — | — | — | — |
| **Methacrolein** | 100 | 0 | 0 | 0 | 100 | 0 | 0 | 0 | 97 | 1 | 0 | 3 |
| **Propenal** | 100 | 0 | 0 | 0 | 100 | 0 | 0 | 0 | — | — | — | — |
| **1,3-butadiene** | 100 | 0 | 0 | 0 | 100 | 0 | 0 | 0 | — | — | — | — |
| **Ethyne** | 100 | 0 | 0 | 0 | 100 | 0 | 0 | 0 | — | — | — | — |
| **Isoprene** | 100 | 0 | 0 | 0 | 100 | 0 | 0 | 0 | 98 | 2 | 0 | 0 |
| **1-pentene** | 100 | 0 | 0 | 0 | 100 | 0 | 0 | 0 | — | — | — | — |
| **1-butene** | 100 | 0 | 0 | 0 | 100 | 0 | 0 | 0 | — | — | — | — |
| **1-propene** | 100 | 0 | 0 | 0 | 100 | 0 | 0 | 0 | 96 | 4 | 0 | 0 |
| **Methyl vinyl ketone** | 99 | 0 | 0 | 1 | 99 | 0 | 0 | 1 | 93 | 4 | 0 | 3 |
| **Furanone** | 100 | 0 | 0 | 0 | 99 | 0 | 1 | 0 | 100 | 0 | 0 | 0 |
| **Styrene** | 100 | 0 | 0 | 0 | 98 | 0 | 2 | 0 | 96 | 3 | 1 | 0 |

*Significant loss by NO3*

| | F26 | | | | F38 | | | | Muller et al. (2016) | | | |
|---|---|---|---|---|---|---|---|---|---|---|---|---|
| | OH | $O_3$ | $NO_3$ | *hv* | OH | $O_3$ | $NO_3$ | *hv* | OH | $O_3$ | $NO_3$ | *hv* |
| **Catechol** | 93 | 0 | 7 | 0 | 66 | 0 | 34 | 0 | 82 | 1 | 17 | 0 |
| **o-cresol** | 98 | 0 | 2 | 0 | 83 | 0 | 17 | 0 | 94 | 0 | 6 | 0 |
| **2,3-dimethyl phenol** | 98 | 0 | 2 | 0 | 84 | 0 | 16 | 0 | — | — | — | — |
| **p-benzoquinone** | 99 | 0 | 1 | 0 | 91 | 0 | 9 | 0 | — | — | — | — |
| **Phenol** | 99 | 0 | 1 | 0 | 88 | 0 | 12 | 0 | 97 | 0 | 3 | 0 |
| **monoterpenes** | 98 | 1 | 1 | 0 | 92 | 1 | 7 | 0 | — | — | — | — |

*Significant loss by photolysis*

| | F26 | | | | F38 | | | | Muller et al. (2016) | | | |
|---|---|---|---|---|---|---|---|---|---|---|---|---|
| | OH | $O_3$ | $NO_3$ | *hv* | OH | $O_3$ | $NO_3$ | *hv* | OH | $O_3$ | $NO_3$ | *hv* |
| **2-furfural** | 0 | 0 | 0 | 100 | 1 | 0 | 0 | 99 | 99 | 0 | 0 | 1 |
| **Acetone** | 18 | 0 | 0 | 82 | 16 | 0 | 0 | 84 | 77 | 0 | 0 | 23 |
| **2,3-butanedione** | 48 | 0 | 0 | 52 | 46 | 0 | 0 | 54 | 1 | 0 | 0 | 99 |
| **Methyl ethyl ketone** | 70 | 0 | 0 | 30 | 68 | 0 | 0 | 32 | — | — | — | — |
| **Benzaldehyde** | 74 | 0 | 0 | 26 | 73 | 0 | 0 | 27 | — | — | — | — |
| **Hydroxyacetone** | 93 | 0 | 0 | 7 | 91 | 0 | 0 | 9 | 88 | 0 | 0 | 12 |
| **Methyl glyoxal** | 93 | 0 | 0 | 7 | 91 | 0 | 0 | 9 | — | — | — | — |
| **Glyoxal** | 95 | 0 | 0 | 5 | 93 | 0 | 0 | 7 | — | — | — | — |
| **Formaldehyde** | 99 | 0 | 0 | 1 | 98 | 0 | 0 | 2 | 39 | 0 | 0 | 61 |
| **Acetaldehyde** | 98 | 0 | 0 | 2 | 98 | 0 | 0 | 2 | 95 | 0 | 0 | 5 |

[Figure]

**Figure S1.** Modeled ozone compared to ozone measured during a dark, low NMOG (< 70 ppb) experiment. Output from the model is shown assuming that the dilution stream contains 60, 70, and 80 ppb of ozone. The input of ozone with the best fit (70 ppb) is applied to the photochemistry model described in Section 2.4

[Figure]

**Figure S2.** Small chamber (A) $NO_x$/NMOG ratio, (B) NMOG composition, (C) total NMOG loading, and (D) total $NO_x$ prior to photochemical oxidation. Panel B shows the fraction of the total NMOG signal attributable to high temperature, low temperature, and duff pyrolysis as defined by Sekimoto et al. (2017). The grey bars indicate experiments in which initial NMOG loadings are sufficiently low to avoid significant OH titration.

[Figure]

**Figure S3.** Distribution of monoterpenes (*m/z* 137) and other select NMOG measured from the combustion of (A) Engelmann spruce, (B) Douglas fir, and (C) subalpine fir using GC-PTR-ToF-MS.

**Furan Reactions**

[Figure]

**Figure S4.** Furan reactions implemented into the MCM box model. Reactions are based on mechanism reported by Aschmann et al. (2014). Names in red indicate species currently represented in MCM v 3.3.1.

**Figure S5.** 2-methylfuran reactions implemented into the MCM box model. Reactions are based on mechanism reported by Aschmann et al. (2014). Names in red indicate species currently represented in MCM v 3.3.1.

**Dimethyl Furan Reactions**

[Figure]

**Figure S6.** 2,5-dimethylfuran reactions implemented into the MCM box model. Reactions are based on mechanism reported by Aschmann et al. (2014). Names in red indicate species currently represented in MCM v 3.3.1.

**Furfural Reactions**

[Figure]

[Figure]

**Figure S7.** Furfural reactions implemented into the MCM box model. Reactions are based on mechanism reported by Zhao and Wang (2017). Names in red indicate species currently represented in MCM v 3.3.1.

**Furfural Reactions (Continued)**

[Figure]

[Figure]

**Figure S8.** Furfural reactions implemented into the MCM box model (continued from Fig. S7). Reactions are based on mechanism reported by Zhao and Wang (2017). Names in red indicate species currently represented in MCM v 3.3.1.

**Methyl Furfural Reactions**

[Figure]

**Figure S9.** 5-methylfurfural reactions implemented into the MCM box model. Products and branching ratios are assumed to follow pathways analogous to the furfural mechanism reported by Zhao and Wang (2017). Names in red indicate species currently represented in MCM v 3.3.1.

**Figure S10.** Guaiacol reactions implemented into the MCM box model. Reactions are based on the guaiacol mechanism reported by Lauraguais et al. (2014). GUAIACOLPROD is a generic, non-reactive product that is included to account for the fraction of guaiacol that does not react to form nitroguaiacols

[Figure]

**Figure S11.** Primary NMOG measurements (blue lines) compared to modeled output (black dotted lines) for Fire 26. Fuel = Englemann Spruce Duff, $NO_x$/NMOG = 0.02, primarily duff pyrolysis NMOG mixture.

[Figure]

**Figure S12.** Secondary NMOG measurements compared to modeled output for Fire 26. Row (A) shows PTR-ToF-MS measurements of $C_4H_2O_3$ compared to model output of maleic anhydride. Row (B) shows I-ToF-CIMS measurements of $C_5H_6O_3$ compared to model output of methyl hydroxy furanone and its tautomer, $\beta$-acetylacrylic acid. Row (C) shows I-ToF-CIMS measurements of $C_4H_4O_3$ compared to model output of hydroxy furanone, its tautomer malealdehydic acid, and 2,3-dioxobutanal. All graphs to the left show full model runs, while graphs to the right show model runs when the initial conditions of furan, 2-methylfuran, 2,5-dimethylfuran, furfural, 5-methylfurfural, and furanone are set to zero

[Figure]

**Figure S13.** PTR-ToF-MS and I$^-$-ToF-CIMS measurements of formaldehyde, acetaldehyde, and nitroguaiacol compared to model ouput for F38.

[Figure]

**Figure S14.** Fate of $RO_2$ species that lead to the formation of hydroxy furanone and methyl hydroxy furanone for F26, F38, and the ambient plume described by Müller et al. (2016). Shown is the fraction of $RO_2$ loss associated with reactions with $HO_2$, NO, and other $RO_2$ radicals.

[Figure]

**Figure S15.** Comparison of NMOG distributions measured in the stack and small chamber prior to OH oxidation for (A) F66 - Sagebrush, (B) F26 - Englemann Spruce Duff, and (C) F29 Chamise. The bottom row shows the NMOG/$C_2H_3N$ ratio for each species measured by the PTR-ToF-MS, along with 1:1, 2:1, and 1:2 lines.

[Figure]

**Figure S16.** Model sensitivity of (A) maleic anhydride, (B) ozone, and (C) hydroxyfuranone to the assumed branching ratios of the furfural mechanism (Fig. 6). The "base case" simulation assumes branching ratios of A = 0.37, B = 0.6, B = 0.03 while the "equal weight" simulation assumes A = 0.33, B = 0.33, C = 0.33. All other simulations assume that furfural loss follows a single channel (i.e., all channel A, B, or C). Model output are compared to the measurements of maleic anhydride and ozone reported by Müller et al. (2016). Measurements of hydroxy furanone are not available.